# Gene-by-environment interactions in urban populations modulate risk phenotypes

Marie-Julie Favé[1,2], Fabien C. Lamaze[1], David Soave[1], Alan Hodgkinson[2,3], Héloïse Gauvin[2,4], Vanessa Bruat[1,2], Jean-Christophe Grenier [1,2], Elias Gbeha[1], Kimberly Skead[1], Audrey Smargiassi[5], Markey Johnson[6], Youssef Idaghdour[7] & Philip Awadalla[1,2,8,9]

Uncovering the interaction between genomes and the environment is a principal challenge of modern genomics and preventive medicine. While theoretical models are well defined, little is known of the G × E interactions in humans. We used an integrative approach to comprehensively assess the interactions between 1.6 million data points, encompassing a range of environmental exposures, health, and gene expression levels, coupled with whole-genome genetic variation. From ~1000 individuals of a founder population in Quebec, we reveal a substantial impact of the environment on the transcriptome and clinical endophenotypes, overpowering that of genetic ancestry. Air pollution impacts gene expression and pathways affecting cardio-metabolic and respiratory traits, when controlling for genetic ancestry. Finally, we capture four expression quantitative trait loci that interact with the environment (air pollution). Our findings demonstrate how the local environment directly affects disease risk phenotypes and that genetic variation, including less common variants, can modulate individual's response to environmental challenges.

[1] Ontario Institute for Cancer Research, Toronto, ON M5G 0A3, Canada. [2] Sainte-Justine Research Center, Faculty of Medicine, University of Montreal, Montreal, QC H3T 1C5, Canada. [3] Department of Medical and Molecular Genetics, Guy's Hospital, King's College London, London, WC2R 2LS, UK. [4] Statistics Canada, Ottawa, ON K1A 0T6, Canada. [5] Department of Environmental Health and Occupational Health, University of Montreal, Montreal, QC H3N 1X9, Canada. [6] Health Canada, Air Health Science Division, Ottawa, ON K1A 0K9, Canada. [7] NYU Abu Dhabi, Abu Dhabi, UAE. [8] Department of Molecular Genetics, University of Toronto, Toronto, ON M5S 1A1, Canada. [9] Ontario Health Study, Ontario Institute for Cancer Research, Toronto, ON M5G 0A3, Canada. Correspondence and requests for materials should be addressed to P.A. (email: Philip.Awadalla@oicr.on.ca)

Environmental exposures, coupled with genetic variation, influence disease susceptibility, and deconstructing their respective contributions remains one of the principal challenges in understanding complex diseases[1–7]. Individuals with different genotypes may respond differently to environmental variation and generate an array of phenotypic landscape[8–14]. Such gene-by-environment interactions are thought to be pervasive and may be responsible for a large fraction of the unexplained variance in heritability and disease risk[9,15,16]. Yet, disease risk, owing to either environmental exposures and/or their interactions with genotype, remains poorly understood[2,17,18].

Canada's precision medicine initiative, the Canadian Partnership for Tomorrow Project (CPTP: http://www.partnershipfortomorrow.ca) is a cohort comprising over 315,000 Canadians, and captures over 700 variables, ranging from longitudinal health information to environmental exposures, to determine genetic and environmental factors contributing to chronic disease. The program includes the Quebec regional cohort, CARTaGENE, which has enrolled over 40,000, predominantly French-Canadian (FC) individuals between 40 and 70 years of age[19–21], to date. Drawing from this founding population of individuals with largely French ancestry, we selected 1007 individuals to determine mechanisms by which genomes, the environment, and their interactions contribute to phenotypic variation. After attributing a regional and/or continental ancestry to each individual using genome-wide polymorphism data, we are able to capture the effect of different environmental exposures on gene expression and health-related traits, while simultaneously controlling for genetic relatedness and migration. Further, in order to capture gene-by-environment interactions through eQTL analyses, we combine whole-transcriptome RNA-Sequencing profiles with whole-genome genotyping and extensive fine-scale environmental exposure data.

## Results

**Population history reveals a fine-grained regional structure.** Individuals selected for analyses include those living across different regions in Quebec: Montreal, the largest urban center in the Quebec province (MTL, 4500 individuals/km$^2$); Quebec City, a smaller urban center (QUE, 1140 ind/km$^2$); and Saguenay-Lac-Saint-Jean, a less urbanized region (SAG, 800 ind/km$^2$). Differences in the regional environment within and across these cities, including ambient pollutant concentrations, are known to be associated with various health outcomes[22,23]. The majority of the Quebec population is of FC descent; a group of individuals descending from French settlers that colonized the Saint-Lawrence Valley from 1608 to the British conquest of 1759[24]. Despite considerable expansion, the population remained linguistically and religiously isolated while remote regions were colonized by small numbers of settlers, such as SAG[25,26] and contributed to the establishment of subpopulations. These sequential population bottlenecks impacted the genome of FCs through increasing the relative deleterious mutations load[27], while reducing overall genetic diversity in the population relative to the European population[28]. Using high-density whole-genome genotyping assays (Illumina Omni 2.5), we confirm that FCs ($n = 689$) form a distinct genetic cluster relative to those of European descent ($n = 136$) (Fig. 1a, Supplementary Fig. 1a–c), as has been previously observed[27]. Within this FC group, we capture fine-scale regional genetic variation across Quebec (Fig. 1c, b and Supplementary Fig. 1d), consistent with Quebec settlement history and local ancestry.

**Ancestry contributes marginally to regulatory variation.** We were particularly interested in the extent to which individual

regional-ancestry and regional-environment account for transcriptional variation in the Quebec population. In an attempt to reduce batch effects in our RNA sequencing experiment, the sampling protocol was standardized across all clinics and all manipulations were performed in the same laboratory. Furthermore, participant's fasting blood samples were collected by CARTaGENE between 9 a.m. and 11 a.m. Individuals were randomized across sequencing lanes to reduce false associations with traits owing to sequencing differences across lanes. Corrections to mitigate remaining batch effects, unwanted technical and biological variation in gene expression were applied (Supplementary Fig. 4)[29] (Methods). Using whole-genome genotyping, we are able to distinguish between "FC-locals" and "FC-regional migrants" (Fig. 1b, Supplementary Fig. 1d, Supplementary Table 2). We define "FC-locals" as individuals of regional ancestry identical to the region they reside in and "FC-regional migrants" as FC immigrants from a different regional ancestry. Among FC-locals, an increasing number of genes are significantly differentially expressed between Mtl- vs Que-locals, $n = 505$, Que- vs Sag-locals $n = 2167$, up to $n = 6649$ and Mtl- vs Sag-locals (Fig. 2a) ($p$ value $< 0.05/15,632$, log-fold change $> 0.5$). Additionally, a greater number of genes are differentially expressed between individuals having the same regional ancestry but who reside in different regions (FC-locals vs FC-regional migrants with the same genetic ancestry, but residing in different regions), and we find this pattern in nearly all pairwise comparisons of this nature (Fig. 2b). On the other hand, when we performed comparisons between FC-locals and FC-regional migrants, we find very few differentially expressed genes in nearly all comparisons (Fig. 2c).

We replicate these findings by performing comparisons of Europeans and FC-locals residing within the same region and find very few differentially expressed genes between them (Fig. 2d, Supplementary Fig. 5). The lack of differentially expressed genes is not attributable to differences in statistical power as we are able to identify up to 75% of our differentially expressed genes using only 30% of our FC individuals ($n = 200$) (Supplementary Fig. 6). Furthermore, results are consistent after performing differential expression analyses between regions using a resampling-based method (1000 replicate permutations for each pairwise comparison between regions), thus reducing the possibility that undetected sampling differences between regions, or outlier individuals, drive those patterns. Differentially expressed genes between regions are enriched for genes implicated in oxygen and gas exchange, G-protein-coupled receptors, and inflammatory response (Supplementary Fig. 7, Supplementary Table 3). Although we initially captured both genotypic and transcriptional variation correlated with geographic structure among the FC subpopulations, these results indicate that shared regional environmental exposures influence peripheral blood expression profiles to a greater extent than regional or local (and continental) ancestry, and point to potential critical exposures contributing to pathways, phenotypic variation, and possibly disease development.

**Environment shapes regulatory profiles and clinical traits.** To test whether environmental exposures contribute to the geographic variation associated with transcriptional profiles and clinically relevant phenotypes across Quebec, a large collection of fine-scale environmental data (Supplementary Fig. 8 and 9, Supplementary Table 4): satellite-land-use regression models (particulate matter 2.5 (PM2.5) and nitrogen dioxide (NO$_2$)), community land-based measures (ozone (O$_3$) and sulfur dioxide (SO$_2$) for air pollution) are collated. Community level estimates of socio-economic indices (social and material deprivation, population density), and built environment features (greenness, food

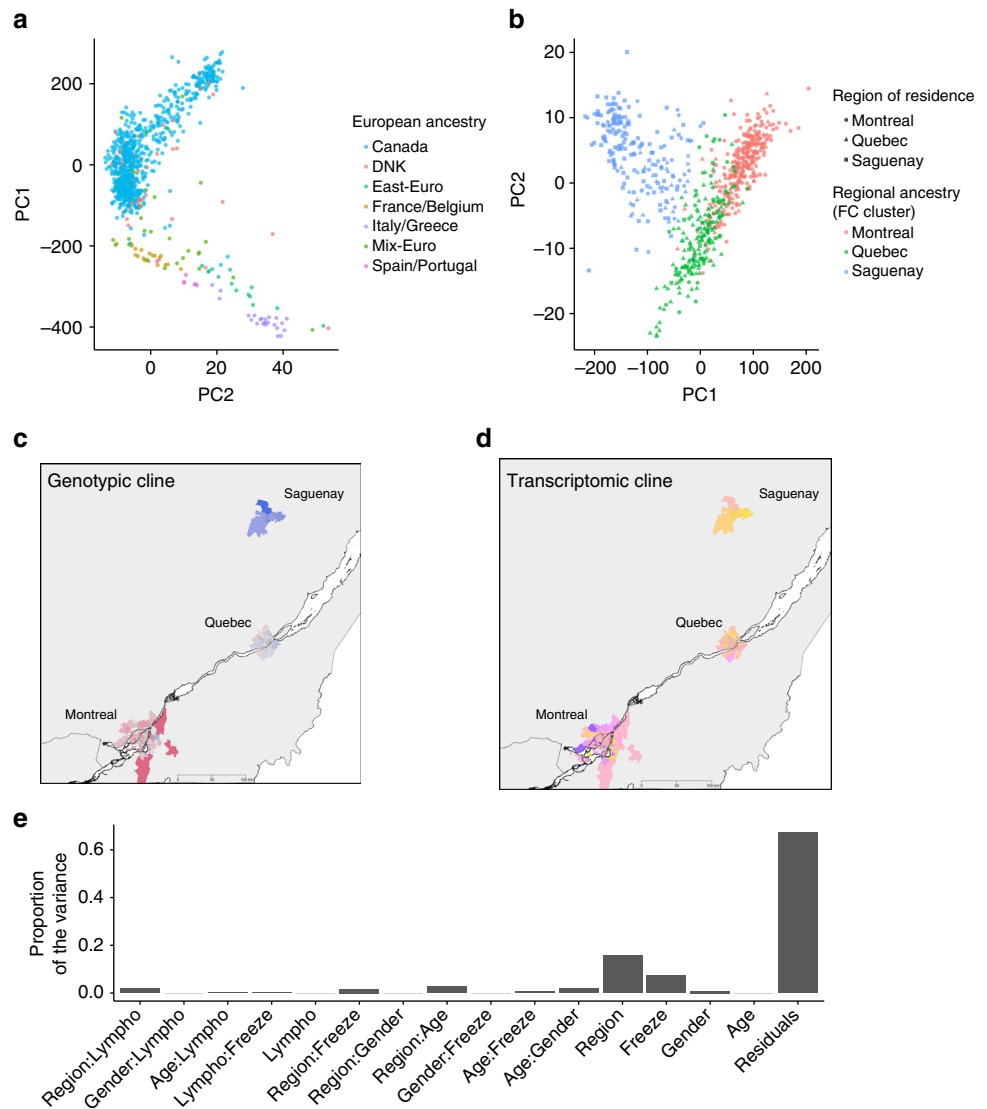

**Fig. 1** Genetic and transcriptomic variation within the CARTaGENE cohort sample. **a** Principal component analysis (PCA) of individuals of European descent, including FCs (n = 887). Individuals are labeled according to self-declared ancestry based on the origin of four grandparents. **b** PCA on the haplotype chunk[61] count matrix of French-Canadians (n = 689) reveals three groups corresponding the region of residence, with SAG individuals showing less overlap with either of MTL or QUE individuals, in line with their historical isolation[25, 26]. **c** Genotypic cline for individuals by location of residence (three-digit postal code) sampled across the province. Color indicate the average value of the first principal component from a PCA on genotypes in each three-digit postal code district level (n = 157). **d** Transcriptomic cline for individuals by location of residence sampled across the province. Colors represent the average value of the first principal component from a PCA on the transcriptome in each three-digit postal code district level (n = 189). **e** Proportion of transcriptomic variance (PVCA) in FCs explained by low-level phenotypes and their interactions

availability, walkability, park density, street network) are also incorporated. A total of 12 environmental exposures, all of them measured or estimated at the level of three-digit postal code (these areas in Canada include several houses or a neighborhood, and their sizes are inversely proportional to population density) (Supplementary Table 4), are included. Indeed, we observe that these environmental exposures capture broad environmental correlates and variance across the Quebec province (Supplementary Fig. 9). In an attempt to pinpoint if particular environmental exposures contribute more to the gene expression differences across regions, we use this fine-scale information for the analytical treatment of individual exposures specific to the individual and ignore broader geographic sampling categories (i.e., regions).

We find that the expression profiles of differentially expressed genes between regions are largely associated with gradients of annual ambient air composition across Quebec (Fig. 3, Supplementary Fig. 9). A north–south urbanization gradient is indeed reflected by higher annual concentrations of PM2.5 and $NO_2$ in downtown Montreal (data derived from satellite-land-used regression models), however, higher concentrations of $SO_2$ and $O_3$, (land-based measures) are observed in SAG (Supplementary Figs. 8 and 9). The higher annual concentrations of $SO_2$ in SAG, a smaller urban center, are related to the presence of several large industrial complexes[22,30]. It is widely known that ambient air pollution covaries with season, and we account for blood collection date in our models (Supplementary Fig. 4c). However, we cannot fully exclude a possible residual contribution of season on gene expression patterns.

We apply coinertia analyses[31] (CoIA) to our multidimensional data to capture associations between 57 clinical endophenotypes (Supplementary Table 5), environmental exposures

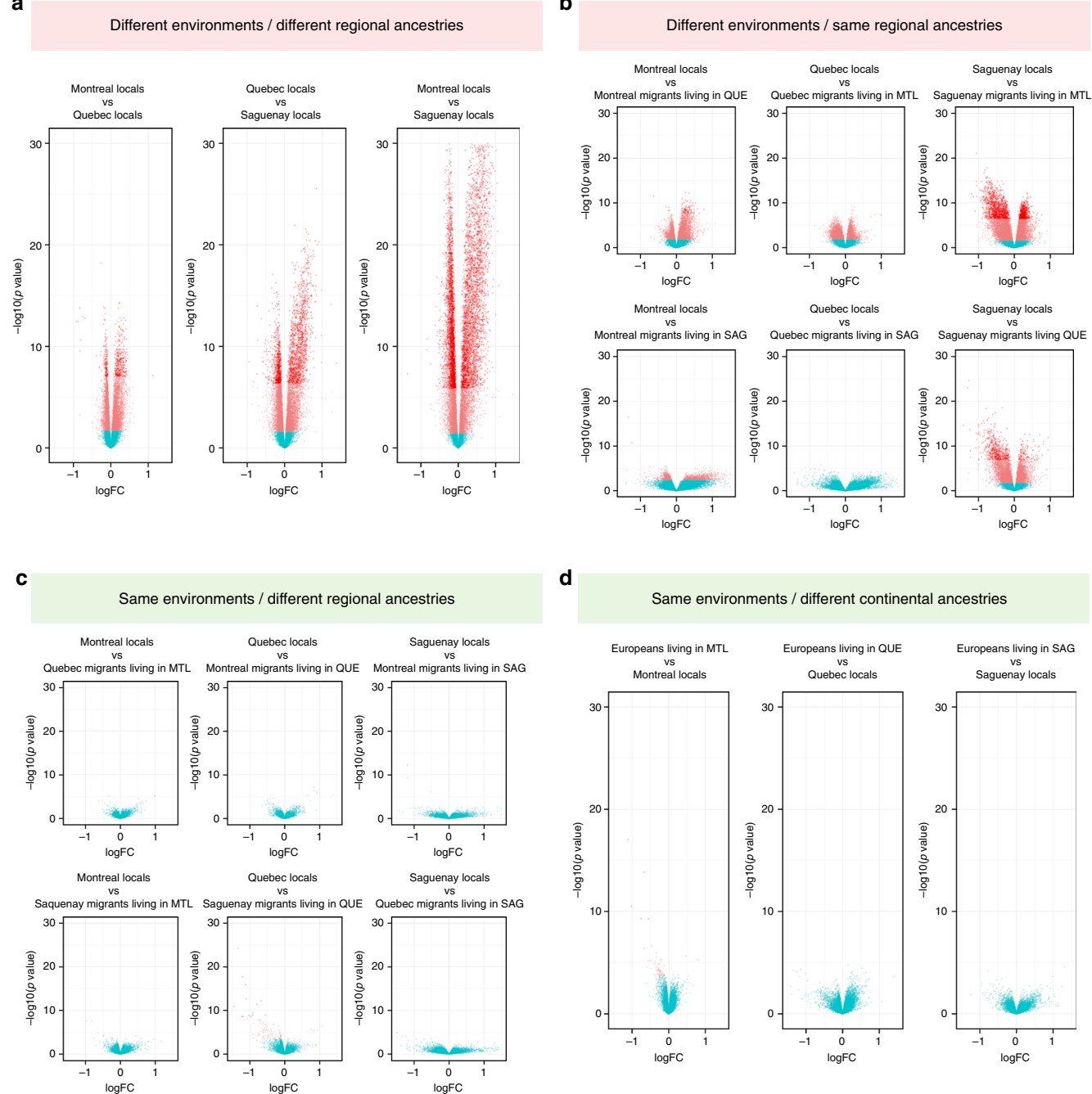

**Fig. 2** Environmental impacts on gene expression profiles override that of genotype. Contrasting the effects of ancestry and regional environment on differential gene expression. **a** Between FC-locals (different regional ancestry, different regional environments). **b** Between FC-locals and FC regional migrants (same regional ancestry, different regional environments). **c** Between FC-locals and FC regional migrants (different regional ancestries, same regional environment). **d** Between FC-locals and Europeans (different continental ancestries, same regional environment). Pink dots are genes with FDR ($q$ value) below 5% and red dots are genes with $p$ value < Bonferroni-corrected $p$ value ($3.20 \times 10^{-6}$)

(Supplementary Fig. 12), and expression levels of differentially expressed genes and their regulators (Supplementary Fig. 10). All phenotypes are standardized health tests captured by CARTa-GENE, and all self-reported disease diagnostics were cross-validated with electronic health records of the participants[19]. Consistent with previously documented effects of air pollution on cardiac and respiratory traits[32,33], we find that arterial stiffness measures, asthma and stroke prevalence, monocytes counts, low-density lipoprotein (LDL), respiratory function (FEV1), as well as liver enzyme levels (Alanine aminotransferase level (ALT),

aspartate aminotransferase level (AST), and gamma-glutamyl transferase (GGT)) show the strongest associations with annual $SO_2$ and $O_3$ ambient levels (Supplementary Fig. 10). In our cohort, the gradient of $SO_2$ exposure is associated with detectable detrimental effects on cardio-respiratory phenotypes, more so than ambient annual PM2.5 and $NO_2$ levels (Supplementary Fig. 10), and is the environmental variable that has the highest replicability of association with gene expression (Fig. 3). As a result of these strong associations of annual $SO_2$ ambient levels with detrimental cardio-respiratory phenotypes, and as $O_3$ levels

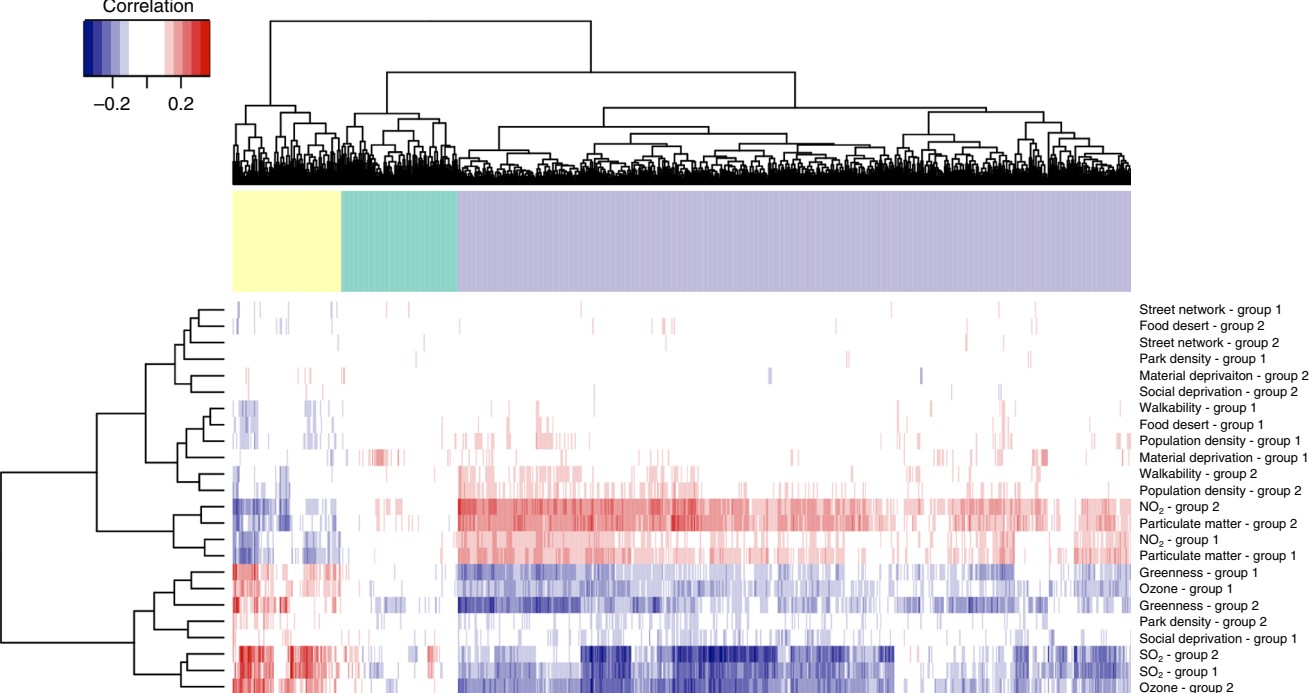

**Fig. 3** Differentially expressed genes are associated with local ambient air pollution. Coinertia (CoIA) analysis between gene expression (columns) and fine-scale environmental variables (rows). CoIA analyses were performed on genes that were significantly differentially expressed among regions and the regulators of those genes (RDEG). CoIAs were computed between differentially expressed genes profiles and fine-scale environmental data (Supplementary Figs 11 and 12). We performed two sets (Group 1 and Group 2, each composed of a random draw of half the cohort) of CoIAs: each set included 10,000× resampling of 200 individuals (without replacement, from Group 1 or Group 2), and the CoIAs were performed between environment and gene expression for each of the 10,000 iterations. Supplementary Fig. 11 depicts the resampling scheme. The heatmap represents, for each Group 1 or Group 2, the median of each environment–gene associations from the cross-tabulated values distribution. Associations from Group 1 and Group 2 largely cluster together, indicating a strong signal of the association between fine-scale air pollution levels and gene expression. A permutation test ($n = 10,000$ steps) indicates the that the correlations between the matrices are significant ($p = 0.00089$ and $p = 9.9 \times 10^{-5}$ for Group 1 and 2 respectively)

are more dependent on other various ambient factors (sunlight, other $NO_x$ emissions), we focus our high-resolution analyses on the participant's weekly $SO_2$ exposure.

We use a 2-week exposure to $SO_2$ pollution, obtained from averaging over a 14-day window preceding the time-point of each individual blood sampling (Supplementary Fig. 14). The large temporal fluctuations in weekly $SO_2$ ambient concentrations allow us to include individuals from SAG that were exposed to low levels of $SO_2$ (despite SAG having high annual averages), and MTL individuals exposed to high levels of $SO_2$ (despite MTL having lower annual averages), or vice-versa. In that way, we can single out the effect of the local environment itself, predominantly attributable to $SO_2$ exposure, to the broader regional effect detected earlier. Using a robust resampling approach to balance the number of individuals in each category, we are able to identify with confidence 170 differentially expressed genes between high- and low-$SO_2$-exposure individuals; these are also found to be differentially expressed between regions (Fig. 2a, Supplementary Table 6).

Furthermore, while multivariate models show that gene expression variation for those 170 genes is significantly associated with 2-week $SO_2$, they do not show an association with smoking, socioeconomic status, or with most built environment characteristics (Supplementary Table 6). We perform a sensitivity analysis using MTL-only samples, thereby removing the potential influences of geographic region and regional ancestry. We replicate these associations with pollution, and the lack thereof, for smoking and socio-economic status (Supplementary Table 6). These results indicate that the regional effect on the gene expression is mostly associated with ambient air pollution, and

less so, or not at all, with diseases, smoking, or the socio-economic factors that were measured. Those 170 differentially expressed genes are again enriched in oxygen-transport activities, and in several pathways involved in leukocyte migration during chronic inflammation, including CXCR chemokine activity and G-protein-coupled receptors (Supplementary Table 6). Circulating blood leukocytes can migrate to sites of tissue injury by responding to proinflammatory cues and are known to migrate through the blood flow to lung epithelial cells during inflammatory response[34].

To disentangle the effects of $SO_2$ exposure from the effects of region on gene expression, we conducted a sensitivity analysis and show that not only is this pattern observed across the whole Quebec province, but it also replicates within Montreal (Supplementary Table 7), suggesting that $SO_2$ exposure, rather than the region itself, is modulating these associations. We find that the expression of the 170 DEGs (between high- and low-exposure to $SO_2$) is also associated with four key clinical traits (Forced expiratory volume (FEV1), lung disease, liver enzymes, and arterial stiffness) (Supplementary Fig. 12). Additionally, when the effects of these four clinical traits are regressed out from gene expression, $SO_2$ exposure remains significantly associated with gene expression (Supplementary Table 7). This suggests that $SO_2$ exposure itself modulates some of the variation in gene expression, and this variation is not only associated with the underlying health status.

The four clinical traits that were found to be associated with differential gene expression (FEV1, lung disease, liver enzymes, and arterial stiffness), are consistently reported as influenced by air pollution by other studies[35–39]. Chronic diseases developing

from these detrimental endophenotypes (asthma and cardiovascular diseases) are well documented to be associated with air pollution levels[12–14,22,37]. Gamma-glutamyltransferase (GGT) has been reported to occur in atherosclerotic plaques[40], is elevated following pollution exposure[41,42], and is predictive in a dose-dependent manner of cardiovascular risk[43]. Interestingly, we find GGT levels to be associated with the differentially expressed genes across SO$_2$ exposure environments, in particular those genes enriched in blood coagulation and platelet regulation (Supplementary Fig. 12), Collectively, these results reveal associations between environmental pollutants, endophenotypic traits, as well as transcript levels, and that the type and direction of associations are consistent with detrimental effects of air pollution, or a correlated variable, on health status.

**Environment modulates the penetrance of genetic variants**. Environmental factors not only directly affect phenotypic variation, but can also modulate associations between segregating genetic variants and phenotypes[1,44,45]. To discover gene-by-environment interactions in both FCs and Europeans, we identify eQTLs for which the effect size is modulated by exposure with one of four ambient air pollutants (env-eQTLs): PM2.5, NO$_2$, O$_3$, and SO$_2$. First, we identify canonical eQTLs using 5,313,384 genotypes and show a high replication for proximal canonical eQTLs (cis-eQTLs) with previously discovered cis-eQTLs (Supplementary Table 8).

To identify gene-by-environment interactions with air pollution (env-eQTLs), we use a randomly generated discovery cohort ($n = 416$) to perform regressions of gene expression levels (eGenes) on cis-SNPs (eSNPs), pollution level, and the interaction between eSNP and pollution (see Supplementary Fig. 15, for a schematic representation of the procedure/design). We use a four-step process that accounts for multiple testing: (1) we compute Bonferroni-corrected $p$ values, adjusting for the number of eSNPs tested for each gene, (2) we retain the lowest Bonferroni-corrected $p$ value for each eGene and transform this set into $q$ values[46] to determine statistical significance (FDR < 0.05, to correct for the 15,632 total genes tested in the cohort). This results in the identification of ten unique significant eSNP–eGene pairs (with nine unique eGenes). (3) We then examine these candidate pairs in our replication cohort ($n = 417$), where four out of the ten pairs are significantly replicated ($q$ value < 0.05) with the same direction of effect in both the discovery and replication cohorts. Last, (4) all four replicated eSNP–eGenes associations (eGenes, $n = 3$; eSNPs, $n = 4$) remain significant using empirical $p$ value estimates through permutations on the combined cohort ($n = 833$ individuals) (Supplementary Table 9).

Following the application of this stringent filtering, we identify and replicate three eGenes (four eSNP–eGene pairs) for which air pollution (either PM2.5, NO$_2$, SO$_2$, or O$_3$) modulates the association between the genotype of at least one eSNP and the eGene expression (Fig. 4, Supplementary Fig. 16, Supplementary Table 9). One eGene, *atad2*, is identified as interacting with both NO$_2$ and SO$_2$ ambient levels. *zp3* is a glycoprotein interacting with proteins in the extracellular space. Interestingly, two eGenes are ATPases with epigenetic activities, regulating chromatin structure (*smarca2*) or assisting in chromatin and histone binding of transcription factors (*atad2*)[47,48].

Among the significant env-eQTLs (FDR $q$ value < 0.05 in discovery and replication cohorts) (Supplementary Fig. 16, Supplementary Table 9), we identify an interaction with NO$_2$ and the SNP–gene pair rs10156534-*smarca2* (Fig. 4a). Furthermore, we find a deletion (chr9: 3,177,272) in an enhancer downstream of *smarca2* that is significant for an interaction with

NO$_2$ levels (Fig. 4c). SMARCA2 protein is part of the large chromatin remodeling complex SNF/SWI (Fig. 4b), and is required for the transcriptional activation of genes repressed by chromatin by mobilizing nucleosomes. The SNF/SWI complex is a tumor-suppressor gene complex and is also required to activate other tumor-suppressor genes. In addition, it has been found to be potentially contributing to a range of inflammatory diseases, including childhood asthma and systolic blood pressure. Interestingly, as discussed above, we find, in CARTaGENE, that spirometry phenotype (FEV1) and arterial stiffness, which are tightly linked to asthma and blood pressure respectively, are associated with differential expression of genes across regional environments. This suggests that environmental differences in air quality may act on the regulation of several genes and pathways and promote pro-inflammatory states which can lead to cardiorespiratory dysfunction.

The eSNP–eGene pair rs62518566-*atad2* is an env-eQTL that interacts with SO$_2$ and NO$_2$ exposition (Supplementary Fig. 16d and e). ATAD2 protein belongs to a large family of ATPases that contains a bromodomain; that is, a protein domain that reads epigenetics marks on chromatin and affects gene regulation[48]. It is a regulator of chromatin dynamics and acts as a co-activator of estrogen and androgen receptors. *atad2* is associated with several human diseases, and serves as a marker of poor prognosis in a variety of different cancers[49,50].

**Variant frequency and the environmental impact on traits**. Allelic frequency has an inverse relationship with phenotypic variation, and, in particular, on eQTLs susceptibility to environmental modifications. First, an inverse relationship between effect sizes on transcript abundances and lead eSNP minor allele frequencies (MAFs) is observed in our cohort (Supplementary Fig. 17). This pattern is consistent with natural selection acting to stabilize gene expression[51–53]. Second, using the estimated correlations from a CoIA analysis between all SNPs in cis of significant eGenes and endophenotypic traits, we test whether the size of the correlations are related to the MAF of the SNP. To do so, we classify the SNPs as common (MAF > 0.1), and less common (MAF between 0.05 and 0.1) and, for each endophenotype, calculate the odds ratios of observing less common variants (compared to common) for stronger endophenotypic associations. We find that less common variants are overrepresented for stronger associations between eSNPs and some endophenotypic traits (Supplementary Fig. 18). More specifically, respiratory (Asthma and FEV1/FVC ratio) and cardiovascular (Stroke, peripheral AIX) phenotypes show larger changes in values in individuals with less common variants at env-eQTL loci (Supplementary Fig. 18). These results suggest that SNP allele frequency is negatively correlated with endophenotypic trait changes when influenced by environmental perturbations, which is coherent with previous studies and theoretical predictions[53,54].

Our findings illustrate that the impact of the geographic region of residence on the blood transcriptome overrides that of ancestry. Moreover, ambient air pollution exposures are likely contributing to this regional effect in Quebec and may explain the differences in some clinical traits among regions such as asthma prevalence. Fortunately, in Quebec, and in many parts of the developed world, air quality has improved since the 1980s[30,55]. However, there has been a sharp increase in anthropogenic pollution levels in many parts of Asia caused by the rapid industrialization and increased use of fossil fuel energies. In the context of global climate change, air pollution and hazardous air quality events are predicted to become more frequent and cause additional morbidity and mortality[23]. More broadly, our work shows how environmental exposures modulate gene expression

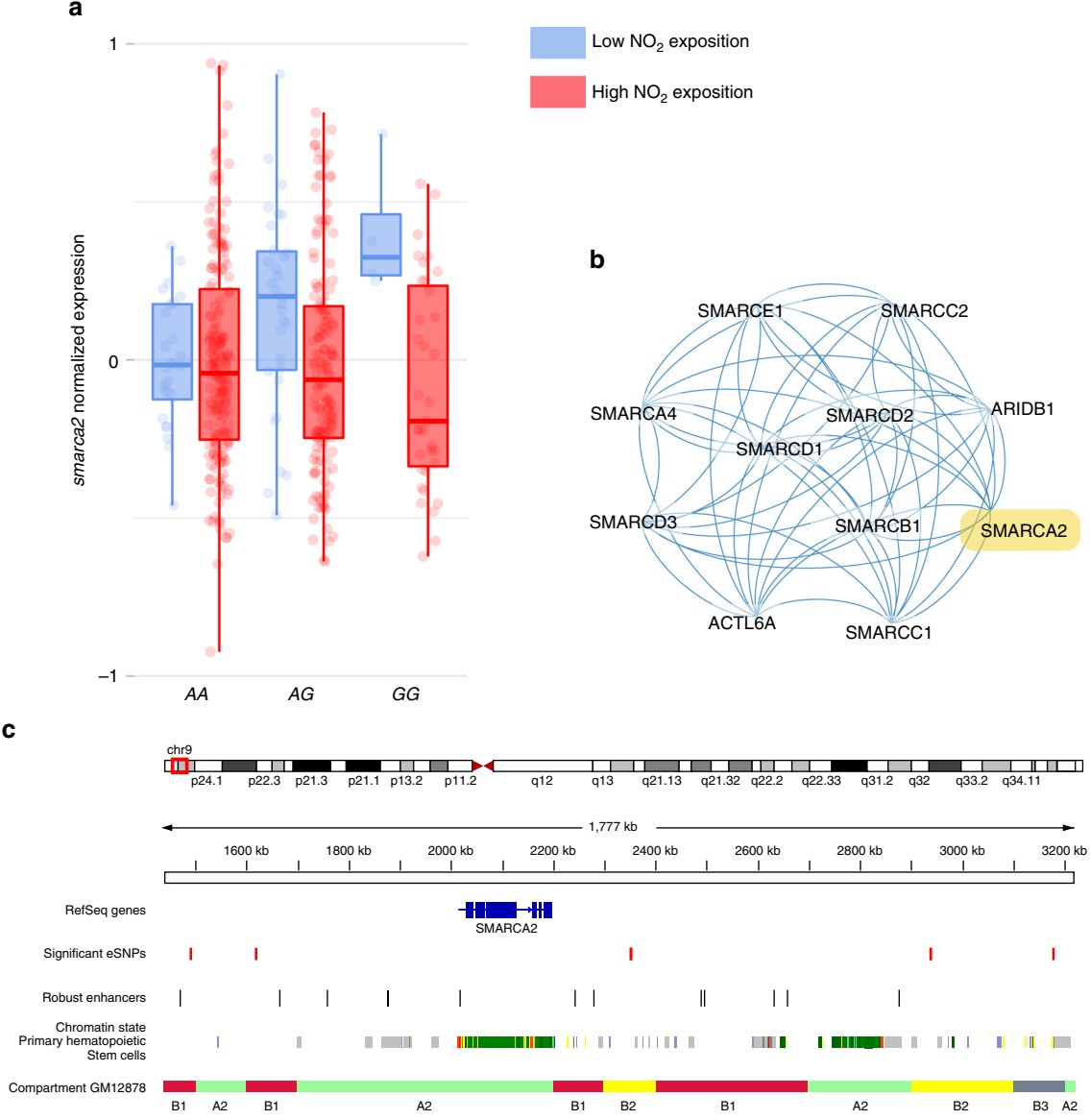

**Fig. 4** $NO_2$ exposure modulates the effect of the top genetic variant rs10156534 on *smarca2* expression. **a** The expression level of *smarca2*, an ATP-dependant helicase involved in several cancers, is modulated by the genotype at rs10156534 and $NO_2$ exposition levels. **b** SMARCA2 is part of a highly connected gene network, the SNF/SWI complex, which acts to remodel chromatin structure and is required to activate transcription of repressed genes. **c** Several enhancers around *smarca2* are found nearby or at location where eSNPs were significant for an interaction with pollution. The upper whiskers extend from the third quartile to the largest value no further than 1.5 * inter-quartile range from the third quartile. The lower whiskers extend from the first quartile to the smallest value at most 1.5 * inter-quartile range from the first quartile

directly, can act upon the penetrance of genetic variants, and can affect clinically relevant phenotypes in humans.

## Methods

**Contact for reagent and resource sharing**. Further information and requests for reagents may be directed to the Biobank CARTaGENE which regulates the access to the data and biological materials (http://www.cartagene.qc.ca/en/contact-us).

**Study population**. The study protocol was approved by the Ethical Review Board Committee of Sainte-Justine Research Center and all participants provided informed consent. CARTaGENE biobank comprises more than 40,000 participants aged between 40 and 60 years, recruited at random among three urban centers in the province of Quebec. CARTaGENE is a regional cohort within the Canadian Partnership for Tomorrow Project, including over 315,000 participants, with various measures obtained from blood parameters, biological function, disease history, lifestyle, and environmental factors[19].

**Sample selection**. For freeze 1, we selected 708 individuals from the CARTaGENE's biobank samples with available Tempus Blood RNA Tubes (ThermoFisher Scientific) and Framingham risk scores, ensuring an equal representation of ages and gender. Two-hundred-and-ninety-two additional samples were subsequently selected from CARTaGENE (freeze 2) based on their RNA and complete arterial stiffness (AIx) measures availability. These samples were selected for having high AIx values as well as average AIx values to complement the first freeze of samples with the intention of achieving a broad range of arterial stiffness values across the complete study cohort. All samples were collected in the same year, with a standardized protocol in all sampling clinics[19]. All blood samples were collected in the morning, on fasting participants.

**Genotyping and QC**. In total, 928 samples with RNA-Seq profiles that passed quality control (QC) thresholds were genotyped on the Illumina Omni2.5 array to obtain high-density SNP genotyping data. A total of 1,213,103 SNP were retained after filtering and QC (Hardy–Weinberg $p$ value > 0.001, MAF > 5% and percent of missing data < 1%).

### Table 1 Summary of *k*-mean clustering

| Pollutant | Low exposure | | High exposure | |
|---|---|---|---|---|
| | Cluster mean by pollutants | Number of individuals | Cluster mean by pollutants | Number of individuals |
| PM2.5 | 8.95 | 392 | 5.97 | 605 |
| NO$_2$ | 5.86 | 160 | 14.34 | 837 |
| O$_3$ | 22.97 | 775 | 25.05 | 222 |
| SO$_2$ | 0.72 | 339 | 1.90 | 658 |

Cluster means and number of individuals within each categories

**RNA sequencing**. Whole blood samples were collected from participants in 2010. Total RNA was isolated using the Tempus Spin RNA isolation kit (ThermoFisher Scientific) and a globin mRNA-depletion was performed using the GLOBINclear-Human kit (ThermoFisher Scientific). The quality and integrity of the RNA samples were verified using an Agilent Bioanalyzer 2100 and all samples had an RNA integrity number (RIN) > 7.5. A RIN above 7.5 is indicative of high quality RNA in the sample and for which RNA degradation is minimal, indicating optimal transport and preservation conditions. Our RIN threshold is more stringent than other large-scale consortium studying gene expression in tissues[51,56]. TruSeq RNA Sample Prep kit v2 (Illumina) was used to construct paired-end RNA-Seq libraries with 500 ng of globin-depleted total RNA. Recommended Illumina protocols were followed for quantification and quality control of RNASeq libraries prior to sequencing. Paired-end RNA sequencing was performed on a HiSeq 2000 platform at the Genome Quebec Innovation Center (Montreal, Canada). Sequencing was performed for freeze 1 (708 samples) using three samples per lane, and for freeze 2 (292 samples) using six samples per lane yielding about 60 million reads per samples. All RNA-seq experimental steps following blood draw were conducted in the same central laboratory, and samples were distributed randomly over sequencing lanes (Supplementary Fig. 3a, b), thereby reducing the introduction of experimental bias at these steps.

Reads were trimmed for adapters and bad quality bases first using Trim Galore and were then assembled to a reference genome (hg19, European Hapmap (CEU) Major Allele release) using STAR (v2.3.1z15)[57] using the two-pass protocol, as recommended by the Broad Institute. The two-pass protocol consists in two consecutive mappings steps having the same set of parameters with only the reference that is optimized in the second mapping procedure. The first mapping is done using the reference gene definition coming from ENSEMBL (release 75). Then, using the splicing junction database files formed by the first pass mapping step for all the samples combined together and the same gene definition file, a second reference is indexed and optimized and is used for the second mapping step. The number of mismatches allowed across pair is five and a soft-clipping step that optimizes alignment scores is also done automatically by STAR. The PCR duplicates were conserved as it was shown that quantification of highly expressed genes were disproportionately affected by PCR duplicates removal[58]. Only properly paired reads were kept (using samtools[59]) for the analysis, according to STAR parameters. After these steps, HTseq (v.0.6.1p1)[60] was launched separately on each alignment file using the same gene reference file that was used for the alignments.

All analyses downstream were conducted using R 3.1.2 and R 3.2.2 and Bioconductor R packages.

**Fine-scale population genetic structure within Quebec**. To unveil finer scale patterns of population structure, i.e., differences between individuals with European ancestry versus individuals having a French Canadian ancestry, we also used ChromoPainter (v0.04)[61], a haplotype-based method powerful enough to detect fine-scale genetic structure. Original genotyping data was used apart from singletons, yielding to 1,908,336 SNPs. Singletons were removed as they are non-informative for phasing and contribute to computation burden for the step of haplotypes sharing inference performed with ChromoPainter. Genotypic data was phased with SHAPEIT (v2.r644)[62] using the HapMap genetic maps. Coancestry matrices were obtained from ChromoPainter with parameters estimation step done with ten iterations on four chromosomes only. ChromoPainter method performs a reconstruction of every individual genome using chunks of DNA donated by the other individuals and report matrices of the number and length of those chunks. We used the chunk count matrix to (1) run FineSTRUCTURE algorithm to build a tree (as recommended for large data set, we performed 10,000,000 burn-in and runtime MCMC iterations) (Supplementary Fig. 1D) and to (2) perform a PCA (Fig. 1a, Supplementary Fig. 1c). Regional ancestry for each FC was determined based on the three clusters obtained from the fineSTRUCTURE tree, (Supplementary Fig. 1d, Fig. 1b).

In agreement with Quebec settlement history, previous studies of the Quebec population[28,63], and the fineSTRUCTURE tree, a PCA of FC individuals reveals groupings of sub-populations of individuals that follow a North–South structure (Fig. 1b, c). The founding event from French settlers followed by the subsequent colonization of remote regions has led to population differentiation among regions in Quebec[28,63]. By further restricting the group of individuals to be analyzed to only FC (n = 726) and considering their region of residence (either Quebec City, Montreal, and Saguenay) a PCA on the chunk count matrix reveals three groups corresponding to region of residence, with the Montreal and Quebec groups overlapping to a greater extent, in line with their greater geographic proximity (Fig. 1b, c). Those three groups were also recovered by the fineSTRUCTURE tree (Supplementary Fig. 1D). Considering all SNPs and the whole haplotypic structure is the key in seeing differences for those two metropolitan regions that have low differentiation. We further identified several participants with a regional ancestry discordant with their region of residence: an indication of recent regional migration of these participants across Quebec regions (Supplementary Table 2).

**Imputation**. To increase the power for the association study with gene expression levels, variant imputation was conducted on 968 individuals for which the genotyping was available from the Illumina Omni2.5 array. We pre-phased the genotypes with SHAPEIT (v2.r64410)[62] using the default parameters, on both the autosomes and the chromosome X. We filtered variants for MAF > 1% and Hardy–Weinberg p value > 0.0001 and passed the haplotypes to IMPUTE2 (v2.2.2)[64] to perform the imputation using the 1000 Genomes Phase I integrated haplotypes (Dec 2013). We used the parameters Ne = 11418 and call thresh = 0.9. We removed variants with a call rate <90%, MAF > 1%, and Hardy–Weinberg p value > 0.0001. A total of 9,157,622 variants passed the filters. Of these, 8,877,297 variants were found on the autosomes and included 779,579 insertion-deletion polymorphisms (indels) (8.78%) and 8,097,718 SNPs (91.22%). 280,325 variants were found on the chromosome X, which included 28,504 indels (10,16%) and 251,821 SNPs (89.84%).

To determine the ancestry of each individual from genotyping data, we carried out a principal component analysis (PCA) with SNPs pruned for LD (pairwise $r^2$ > 0.2 and 50 SNPs window shifting every five SNPs) (Supplementary Fig. 1A), yielding 146,689 SNPs. The continental ancestry (African/European/Asian/Canadian/American/Middle-Eastern) of each individual was determined based on the PCA plot (Supplementary Fig. 1A) and verified as to whether it corresponds to self-reported ancestry based on the country of origin of four grandparents. If the country of origin of three out of four grandparents and the PCA continental grouping were concordant, the individual was assigned to a continental origin.

**RNA-sequencing filtering**. Genes with counts-per-million below 0.5 in more than half of the cohort (505 individuals) were removed from the analysis for a total of 15,632 genes retained for all downstream analyses. Individuals that showed obvious outlier after visual inspection of principal component plots were removed (three individuals).

**Variables contributing to transcriptomic variation**. The deep phenotyping of the CARTaGENE cohort allow for a thorough exploration of the biological and environmental factors that may influence genome-wide gene expression patterns. As most statistical procedures assume a normal distribution to the underlying data, we transformed the normalized counts from freeze 1 to a Gaussian distribution using a log2cpm transformation using edgeR. We summarize the gene expression levels by performing a PCA on the normalized expression matrix (ePCA). To identify variables that contribute to genome-wide gene expression variation, we performed a stepwise regression (stepwise search from both directions) on ePC1 and ePC2. Results of the stepwise regression are given in Supplementary Table S1, as well as the results from the replication analyses using freeze 2. We included the following low level endophenotypes in the stepwise procedure: set, region of residence, cell counts (lymphocytes, neutrophils, monocytes), arterial stiffness, age, and sex.

Using the freeze 1 data set of 708 individuals, we quantified the proportion of the variance in expression attributable to cell counts, age, sex, region, and arterial stiffness (Supplemental material) by using principal variance component analysis (PVCA), and found that the region of residence explains ~16% of the variance in gene expression, while the effects of age, sex, and cell counts were much lower (Fig. 1e). These analyses were repeated on an additional 289 participants (freeze 2) and both of these effects were found to be replicated on expression profiles (Supplementary Table 1). Similarly, when combining transcriptional profiles for all individuals, we found that the region of residence explains ~15% of the variance in gene expression both in FCs and in Europeans (Supplementary Fig. 2).

**Sampling site effect within region**. The RNA extractions and library preparation were performed for all individuals in the same laboratory to reduce technical bias. However, participants were sampled across four different sampling sites inevitably situated within geographical regions where participants lived. Our experimental design was built in such a way that sequencing run was not correlated with region of residence (Supplementary Fig. 3a). To evaluate whether the sampling site has any effect on the RNA-Seq quantification data, we performed extensive analyses of the two sampling sites situated within Quebec City: St-Sacrement (STS, n = 136) and Enfant-Jesus (EF, n = 129). QUE individuals expression profiles from the combined data set show that individuals from STS and EF form a single cluster on a ePCA plot (Supplementary Fig. 3b). Furthermore, a variance component analysis (PVCA) was performed on the QUE individuals only and including sampling site

as an explanatory variable shows that the sampling site explains <5% of the variance within QUE region, while freeze explains 15%, age 5%, and gender 2.5% (Supplementary Fig. 3c). In comparison, in FCs or Europeans, region of residence accounts for 15% of variance in gene expression. In addition, we performed a differential expression analysis between sites within a region (see details below) using permutations, and found that there are no genes differentially expressed between clinics within a region, supporting the absence of sampling differences between clinics affecting gene expression to a detectable and significant level.

**Correction for technical and biological unwanted variation.** High quality RNA-sequencing of all 997 individuals reveals a similar geographic structure in transcriptional profiling than population structure from genotyping (Fig. 1c). Investigation of the variance associated to gene expression reveals that region of residency (variable of interest) explains about 16% (Supplementary Fig. 2a) of the variance regarding the population of origin (Supplementary Fig. 2b, c), but unwanted variables explain a certain proportion of the variance (Fig. 1e, Supplementary Fig. 2b, c).

RNA-Seq data generation, and expression data in general, are prone to technical biases which in some cases can mimic, or be confounded, with biological variation. The appropriate normalization pipeline in an RNA-Seq experiment will depend on the experimental design and the hypothesis being tested. Local sequence context can bias the uniformity of read counts along the genome, and sophisticated normalization pipeline may be necessary when comparing expression levels across genes[65]. Most experimental designs of RNA-Seq studies, like the one presented here, compares different groups of individuals to each other, therefore the normalization pipeline should rather focus on removing unwanted variation across individuals.

We removed the effects of hidden covariates potentially affecting expression levels using surrogate variable analysis (SVA)[29]. We used the SVA correction, retaining five surrogate variable, for the differential expression analyses, correcting for technical (i.e., runs, sets, number of reads) and biological (i.e., date of appointment, time of the year, sex, smoking status, cell counts) effects on gene expression (Supplementary Fig. 4). We performed the same stepwise regression approach as previously, but on the SVA corrected expression level matrices and show that we retained the variation associated with region, but removed any effects of cell counts and arterial stiffness that was present in the uncorrected expression levels (Supplementary Fig. 4, Supplementary Table 1). The corrections do not fully compensate for the effect of the freeze (technical), we therefore include this covariate in all subsequent analyses. Estimating the variance associated with hidden batch has been shown to remove variation associated with biological and technical factors and also increase the power to identify eQTLs[58,66].

**Differential expression analysis.** Because of the large proportion of the variance in gene expression explained by region of residence, we identified genes that are differentially expressed between pairwise comparisons between the FC-locals from the three regions (Montreal, Quebec, and Saguenay). Using edgeR[67], we performed a differential gene expression analysis using the 15,632 genes that passed the QC filters established above. We performed the differential expression modeling using the following statistical model:

$$\mu_{ig} = \beta_g Rr_i + \beta_g Ro_i + B_i + S_g + \epsilon_{ig}$$

where Rr is the region of residence, Ro the region of origin, $B$ is the surrogate variable, representing the batch effect estimated by SVA, and $S$ represent the freeze effect that is included in the final (see below for further details).

The significance level of the test was estimated as a gene $p$ value below the Bonferroni-corrected threshold of $3.20 \times 10^{-6}$ (0.05/15,632). The SVA corrected expression levels retained the variation associated with region, but removed any effects of cell counts that was present in the uncorrected expression levels (Supplementary Table 1).

We performed a power analysis of our ability to detect differentially expressed genes with smaller samples sizes. Several of our comparisons of regional- or continental-migrants with FC-locals involve smaller number of individuals (Supplementary Table 2). We therefore assessed our ability to detect differentially expressed genes by performing differential expression analyses between groups for which we found large number of differentially expressed genes, but using a smaller subset of random individuals (without replacement) of each of these groups. We randomly selected 15 Mtl-locals and 15 Sag-locals, and performed the DGE analysis using the same model as above. We also performed the analysis using 50 Mtl-locals and 50 Sag-locals. In each case, we could identify differentially expressed genes which largely overlap with the differentially expressed genes detected in comparisons using all individuals (Supplementary Fig. 6). We observe that with an increasing number of individuals, our power to detect differentially expressed genes increases and that the identity of the differentially expressed genes detected in each of these comparisons largely overlap (Supplementary Fig. 6).

We further support the effect of region of residency on gene expression by performing differential gene expression analysis across regions using permutations that are even more robust to batch effects. The permutation-DGE analyses confirm that differences are the greatest between MTL and SAG. Similar permutation analyses also show that individuals living in the same region but sampled in different clinics have similar gene expression profiles (Supplementary Fig. 3B),

supporting the absence, if not minor, of effects of sampling procedures on the gene expression across sampling clinics.

**Regional environmental effects on gene expression.** We take advantage of the presence of individuals from different regional and continental origins in our cohort to disentangle further the effects of the genetic background and environmental influences on genome-wide gene expression. We first selected individuals of either FC and European continental ethnicity (Fig. 1a, Supplementary Fig. 1). A total of 798 individuals including 136 Europeans and 662 FC were selected for downstream analyses. We stratified the individuals according to their continental origin (FC vs Europeans), and further stratified the FCs into their assigned genetic ancestry (MTL, QUE, SAG) obtained from the fineSTRUCTURE analysis (Fig. 1b, Supplementary Fig. 1D). We then determined their region of residence (MTL, QUE, SAG) for a total of 12 ancestry-residence groups: we identified individuals for which their origin (Continental or regional) is discordant with the region they reside, which we refer to as continental- and regional-migrants respectively (Supplementary Table 2). We also identified FC individuals for which their regional origin is concordant to the region they reside, which we refer to as FC-locals (Mtl-FC-locals, Que-FC-locals and Sag-FC-locals). We performed the differential gene expression analysis pipeline as described above for different pairs of continental-migrants, regional-migrants, and FC-locals to disentangle the effects of the genetic background and the regional environment on genome-wide expression (Fig. 2). We selected 6649 genes that show differential expression ($p$ value < $3.20 \times 10^{-6}$) in the comparison between Mtl-FC-locals and Sag-FC-locals. Using the 12 origin-living groups and the 6649 genes, we performed an unsupervised clustering and visualized the groupings using a heatmap (Supplementary Fig. 5).

**Gene enrichment and reactome analyses.** Gene enrichment analyses were performed using the topGO package in R, with a Fisher exact test. Differentially expressed genes between MTL-locals and SAG-locals were compared against the 15,632 genes expressed in the CARTaGENE cohort that were retained after QC filters (background). Reactome enrichment analyses were conducted with R the package reactomePA, and here again, the background set of genes was defined as the 15,632 genes expressed in blood that pass our filters (Supplementary Fig. 7 and Supplementary Table 3).

**Fine-scale environmental data.** We obtained air quality measures in the year of sampling (2010) from either land-based stations ($SO_2$, ozone) or national LUR models estimates (PM 2.5 and $NO_2$) incorporating information from land use data and satellite remote sensing[55,68–70]. Built environment variables (street network, population density, food deserts, greenness, walkability) and social and material deprivation indicators were accessed through the Quebec government data portal (https://www.inspq.qc.ca/environnement-bati). All environmental data sources are described in Supplementary Table 4.

Environmental data was available at the three-digit postal code district level (i.e., Forward Sortation Area, FSA), or was reformatted to this geographic scale. Postal code districts in Canada are small geographic areas which assist in delivering mail. Postal codes are a series of six digits that identify a small geographic area in a municipality, usually grouping just a few houses together or a small neighborhood. Three-level digits are larger areas that include several houses, a small neighborhood, or a small village. The population of FSAs in Canada range from a few hundreds to tens of thousands of individuals. Three-digit postal code districts can be of different areas, and are smaller in densely populated areas, and larger in areas of low population density. Maps in Fig. 1c, d and Supplementary Fig. 9 depict three-digit postal code districts as thin gray lines areas, and each district is colored with the mean value of interest in each map. Each individual in the CARTaGENE cohort has a three-digit postal code district associated to it, referring to the location of its primary residence. We assigned fine-scale environmental measures to each individual based on its three-digit postal code.

**Coinertia analyses.** Coinertia analysis (CoIA)[31,71] is a multivariate statistical part of the large family of ordination methods, such as PCA, redundancy analysis (RDA), or canonical correlation analysis (CCA). CoIA is a general approach and existing methods such as the ones mentioned above appear as special cases of it[31]. These methods have been widely used in ecological research, including CoIA which has been more recently developed. Collectively, these methods allow for detecting an underlying data structure between two data tables. CoIA uses a combination of PCA and multivariate linear regressions to detect linear combinations of variables from one data table that explain the variance in the second data table. CoIA is more flexible than RDA or CCA, and overcomes their limitations by allowing for more variables than the number of samples to be tested[31,71], which is generally the case in genome-wide scale analyses (i.e., more genes than individuals). This makes CoIA a method of choice to integrate data of diverse types, and of high-throughput like most omics data.

We first used CoIA analysis to reveal the common structure between differentially expressed genes (Fig. 3, Supplementary Fig. 11) and the fine-scale environmental data. We produced two separate principal component analyses (PCAs) based on continuous encoded matrices of both environmental and gene expression levels (normalized for library size and sequencing freeze). The data were

centered and reduced to one unit of variance prior performing the PCA analysis. We conserved components for each PCA to explain 80% of the variance in the data. We imputed missing data only for the fine-scale environmental data set (there were no missing data in the gene expression matrix) using the function *imputePCA* from the R package missMDA. The coinertia analysis performs a double inertia analysis of each data set and then project the variables of the original environmental and gene expression data sets on the new co-inertia axes. Relationships between the two matrices were assessed by comparing the CoIA estimated from the real data set with the CoIA distribution estimated after bootstrapping. Two sets of 500 of CoIAs were computed independently between gene expression and fine-scale environmental data. Supplementary Fig. 11 depicts the resampling scheme. For each Group 1 or Group 2 ($n = 497$ for each) a total of $10,000 \times$ resampling of 200 individuals (without replacement) were performed. We performed a CoIA for each resampling step. We report the median value of the distribution of the distribution of each environment–gene pair cross-tabulated values for each group. Gene enrichment were performed using gProfiler[72], and using the 15,632 expressed genes that passed our filters in whole blood as the background gene set (Supplementary Table 3). We evaluated the significance of the correlations between the two matrices with a multivariate generalization of the Pearson correlation coefficient (RV coefficient) using a permutation test (RV-test) with 10,000 steps from the R package ade4.

To identify clinically relevant endophenotypes that are associated with fine-scale environmental data, we performed a CoIA between 57 clinically relevant endophenotypes (Supplementary Fig. 10) and fine-scale environmental data. The 57 clinically relevant endophenotypes were selected to encompass physical measures (BMI, height, age, sex), most systems relevant to the human health (cardiovascular system, pulmonary functions, hepatic system, renal system, disease history, vision, immune system) and lifestyle measures (smoking status, alcohol consumption, nutrition, physical activity). All biochemical endophenotypes were measured in a single central laboratory. We resampled 10,000 times 493 individuals from the cohort, and performed CoIA at each step between endophenotypes and fine-scale environmental variables. We report the median value of the distribution of each environment–endophenotype pair cross-tabulated values (Supplementary Fig. 10).

To reveal possible associations between expression levels and endophenotypes, we then performed CoIAs with a similar resampling scheme between 12 selected endophenotypes that were the most strongly associated with air pollutants from Supplementary Fig. 10 (Stroke, Arterial stiffness measures, spirometry measures, Asthma, monocyte counts, LDL, AST, ALT, GGT) and differentially expressed genes (results shown in Supplementary Fig. 12).

**Exposure windows of weekly SO₂.** To increase our resolution in air pollution exposures, we used daily SO₂ ambient levels measured in each three-digit postal code. We calculated the average exposure during the 14 days preceding the blood draw for each participant. This way, we reduce the effect of random fluctuations due to technical artifacts or short-term meteorological anomalies that may affect measurements. Also, changes in gene expression and biomarkers in blood following a pollution exposure has been documented as a relatively fast phenomenon, occurring after just a few days of exposure[36]. We then categorized the participants using a $k$-means algorithm[73] into high exposure or low exposure categories (see details on the number of participants and cluster centers in the eQTL section below).

**DGE between high- and low-SO₂ exposure.** To find differentially expressed genes between high and low exposure individuals, we used the same approach as described above for identifying differentially expressed genes between regions, with the following modifications: given the unbalanced number of individuals in each category (108 high exposure vs 800 low exposure) of exposure, we resampled 100 times 108 individuals with replacement from the low- and high-exposure category and performed the DGE pipeline. We performed the SVA while retaining variation associated with SO₂ exposure. We combined the results of DGE analysis in a list of 468 differentially expressed genes, and from these candidates, 170 genes were also identified as differentially expressed between regions (Fig. 2a). Those strong 170 candidates were used for enrichment, CoIA, and multivariate models. We also identified genes (transcription factors) that regulate our 170 differentially expressed genes (RDEGs) using cytoscape, and we used them in addition to the differentially expressed genes in the CoIA analyses.

**Multivariate models for SO₂ exposure.** In an effort to characterize the effects of confounding variables on pollution exposure, we applied multivariate models on gene expression levels. First, similar as in the differential gene expression analysis, we performed a SVA to remove unwanted variation of technical or unknown biological variables while retaining the variation around SO₂ exposure. We then built multivariate models using the SO₂, O₃, and PM2.5 14-day exposures, as well as the remaining 9 non-pollution environmental exposures (Supplementary Fig. 9), as well as smoking status. Smoking status may indeed cause similar changes in endophenotypes as pollution exposure. We then selected the endophenotypes

revealed by the CoIA as being the most associated with region and pollution exposure (Lung disease, Asthma, Stroke, monocyte counts, liver enzymes (AST, ALT, GGT), Arterial stiffness, spirometry tests, and lymphocyte counts), and tested whether any of these would explain variation in the 170 candidate genes. Furthermore, after having identified the health endophenotypes that are associated with gene expression in MTL and in the whole data set (FEV1, liver enzymes, lung diseases, and arterial stiffness, see Supplementary Figs. 10 and 12), we regressed out their effect from the expression of the 170 candidate genes, and run the multivariate models to test for the effects of environmental variables itself (results collated in Supplementary Table 7).

**env-eQTL analysis.** Environmental factors not only directly affect phenotypic variation, but can also modulate associations between segregating genetic variants and phenotypes[1,44,45]. To discover gene-by-environment interactions, we identified eQTLs for which the effect size is modulated by environmental exposure to one of four ambient air pollutants (env-eQTLs): PM2.5, NO₂, O₃, and SO₂. We categorized the participants using a $k$-means algorithm[73] into two categories, high or low exposure, irrespective of the pollutant type (Table 1). A $k$-means algorithm attempts to partition the individuals into $k$ groups (here, $k = 2$), such that the sum of squared Euclidean distances from points to the assigned centroid (cluster mean) is minimized.

We adopted a strategy (Supplementary Fig. 15) to randomly divide the CARTaGENE cohort into discovery and replication cohorts. During this process, for the discovery of eSNP–eGene pairs, we scan the genome at ± 500 kb of the TSS of gene to find all putative cis-eSNPs. We used the following model where gene expression (Y) is regressed on a given SNP (S), a given environmental air pollutant (E) and the interaction between S and E:

$$\text{Model}: Y_{ijk} \sim S_{ijk} + E_{ijk} + S_{ijk}E_{ijk}$$

The gene expression level, was normalized using an inverse normal transformation, and corrected for relatedness and other batch effects using the SVA R package (see above for further information). Here, we focussed solely on the $p$ value associated with the Student's $t$-statistic for the interaction term $S_{ijk}E_{ijk}$. We applied a Bonferroni correction to the interaction $p$ values for SNP-wise multiple testing within gene and retained the most significant putative eSNP–eGene pair from each gene. We then assessed this set of "best" eSNP–eGene $p$ values for significance across all 15,632 genes at the false discovery rate (FDR) threshold of 0.05 by transforming the set into $q$ values[46]). This represented the set of significant discovery eSNP–eGene pairs to be tested in the replication set. We then reported the environmental eSNP–eGene pairs that were significant (replicated) in the replication cohort ($q$ value < 0.05, adjusted for the ten pairs being tested) and had the same direction of effect in both cohorts ($n = 4$ out of the 10).

To provide support for the replicated environmental eSNP–eGene pairs that we reported as significant, we estimated "honest" empirical $p$ values for the whole sample (discovery + replication) using permutation: for each eSNP–eGene pair we performed the same eQTL modeling ($Y_{ijk} \sim S_{ijk} + E_{ijk} + S_{ijk}E_{ijk}$) and permuted the expression values (Y) before obtaining the test statistic (Student's $t$) for the interaction term. By repeating this procedure 1000 times for each eSNP–eGene pair, we built null distributions to assess the original observed (not permuted) $t$-statistics. The empirical permutation $p$ value for each eSNP–eGene pair was taken as the proportion of permutation $t$-statistics larger than the observed $t$-statistic (Supplementary Fig. 16, Supplementary Table 9).

**Impact of lower frequency variants.** We performed a CoIA analysis between all eSNPs of significant eGenes and endophenotypic traits. To do so, we resampled 1000 times, without replacement, 420 individuals from the cohort, and performed a CoIA at each step between endophenotypes showing variation across environments and the eSNPs. The median value for each endophenotype-eSNP correlation from the 1000 CoIA was calculated. The CoIA results are the correlations between eSNPs and the endophenotypic traits values. We then tested whether the strength of these correlations between eSNPs and endophenotypic traits were related to the MAF of the eSNP by examining the odds ratio of observing less common variants (MAF between 0.05 and 0.1, compared to common variants of MAF > 0.1) for stronger endophenotypic associations (Supplementary Fig. 18). The MAF was estimated from the complete cohort data.

**Data availability.** Genotyping, expression, health phenotypes, and exposure data used in this study are available from CARTaGENE (www.cartagene.qc.ca) or the CPTP portal (http://portal.partnershipfortomorrow.ca) upon request. The built environment data set is publicly available from the Quebec government data portal. The air pollution data set is available upon request to Air Health Effects division, Government of Canada. All environmental data sources are detailed in Supplementary Table 4.

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

## Acknowledgements

We thank the Awadalla Lab for comments on the manuscript, as well as Paul C. Boutros and Veronica Y. Sabelnykova from OICR. We acknowledge financial support from Fond de Recherche du Québec—Santé (FRQS), Genome Quebec, Fonds de Recherche du Québec—Nature et Technologies (FQRNT), Canadian Foundation of Innovation, Ontario Ministry of Research and Innovation Principal Investigator Award (P.A.) and a Canadian Institute of Health Research award (#EC3-144623) to P.A. M.-J.F. is a CIHR Neuroinflammation Postdoctoral Fellow. F.C.L. is a FRQS Postdoctoral Fellow. A.H. is a FRQS Postdoctoral Fellow and currently holds an eMedLab Career Development Fellowship as part of the Medical Bioinformatics Initiative funded by the Medical Research Council, UK (grant number MR/L016311/1). Requests for data published here should be submitted to access@cartagene.qc.ca citing this study.

## Author contributions

M.-J.F., Y.I., and P.A. conceptualized the study. E.G. and Y.I. performed the experimental procedures for sequencing and genotyping. M.-J.F., A.H., J.-C.G., M.J., A.S., and V.B. prepared the data and performed quality control. M.-J.F., F.C.L., D.S., V.B., J.-C.G., and H.G. performed bioinformatics and statistical analyses. M.-J.F., F.C.L., D.S., K.S., V.B., and P.A. wrote and revised the manuscript.

## Additional information

**Competing interests:** The authors declare no competing financial interests.

