## [Peer Review File · Nature Communications]

Reviewers' comments:

Reviewer #1 (Remarks to the Author):

This paper investigates the link between urban environment, gene expression, and genotype in a Canadian cohort. The subject is of high interest and has clear links to important public health concerns. While the results investigating differential expression between the three geographic locations is convincing and well described, I have serious concerns regarding the results presented in the latter half of the paper describing GxE interactions on gene expression. While the differential expression analysis is of interest both generally and to the transcriptomic field, the GxE analysis does not meet publication standards and needs considerable further investigation.

Major comments.

1) The inclusion of locals and internal migrants in the Region DEG analysis is nicely done and adds robustness to the conclusions that the gene expression differences due to location are more significant than due to ancestry. However, the authors should provide some caveats that there may be undetected sampling differences between the regions that could drive the changes in gene expression between the two regions. Were the samples all collected in the same year? With the same protocol at the same time of day used for collection at all locations, etc? In particular, time of year (seasonality) has a large association with gene expression in blood (Dopico, Nat Comm 2015). The authors should include time of year in their list of covariates as the sampling time in the different city seems to vary considerably– there are no collections in autumn in Seguenay for example (Figure S14). Further, air pollution in general is known to vary by time of year, and SO₂ exposure (via visual inspection of Figure S14) appears to be highest in April in all three regions.

2) The authors provide a plot of the SO₂ values by region and by month of collection (Figure S14). They should add similar supplemental figures of the remaining three pollution measurements and state what QC has been done on the pollution measurements.

3) The authors binned the ambient pollution levels into two or three categories for the analyses. The authors should explain why they used categories rather than a continuous trait, and how they chose the cutoffs for the categories, as it appears their cutoffs led to very small sample sizes in some categories. SO₂ was categorized into two categories for DEG analysis, but 3 for GxE analysis. Why the difference?

4) The set of variants used in the eQTL analysis is unclear. In the eQTL section, it is stated that SNPs were filtered to MAF > 5%, and similarly in the section "Genotyping, ethnicity and regional origin of French Canadians" a MAF filter of 5% is also stated. However the text discusses discovery at rare variants, which are usually defined as MAF < 0.1%. Even if the method MAF filters are mis-stated, the genotyping methods state that SNPs with MAF < 1% were filtered prior to imputation. Are all rare calls then based on imputed SNPs? This is surprising as rare variants are difficult to impute and imputed rare variants are generally considered unreliable, especially if imputed off the 1KG panel rather than a larger panel such as HRC. What info score were the imputed SNPs thresholded at? Have the imputed rare-variants been validated with sequencing or SNP typing data in this population? The authors make strong claims about the impact of rare variants on expression, these need to be supported by evidence of the accuracy of their rare variant imputation.

5) I have serious concerns regarding the significance of the GxE analysis. While the authors have used a standard method for detecting GxE on expression (matrix-eQTL cross-linear function), the testing

strategy and in particular, the correction for multiple testing and the replication is not well described and appears inadequate. The GxE study design is unclear – were all genes tested for an interaction or just the ones with canonical eQTLs? It is stated that a gene-specific bonferonni correction was applied. This is unclear – were the results at each gene corrected for the number of variants tested at that gene, or the independent variants? Either way, just correcting for the number of tests at a gene is not sufficient, the authors need to correct for both the total number of genes tested as well as the number of variants. Given the high possibility of outliers driving spurious interactions (especially at rare variants), the authors should also assess whether their interactions withstand permutations. The authors do not provide evidence of any GxE Pvalues, either uncorrected or corrected in the manuscript, these should be reported.

6) The replication of GxE interactions is unconvincing and in fact seems to show that there is no significant replication, suggesting the GxE results are likely false positives. Taking N02 as an example, the authors find 11/683 of their hits in FC are present in EURO, a replication rate of 1.6%. Can the authors assess whether this is different than the overlap expected by chance? Have the authors accounted for the number of replication tests performed in the overlap (ie what p value threshold was used in the replication set? 0.05/number of signals taken into replication?) The authors should also confirm that the same SNP-Gene pair were tested in the discovery and replication set and that a consistent direction of effect was required between the two datasets, as these should be required for replication

7) The authors highlight a few genes linked to interesting biology, but do not state if these genes, PAX5 and AFAP1 and those highlighted in Figure S16 replicated in the two populations. The authors should clearly state if these signals replicated, and provide the p values and plots from both groups (FCs and Europeans) in figure 4 and figure S16.

8) The authors should include plots of all putative replicated env-eQTLs, similar to Figure S16, including the results from both datasets, pvalues and effect size estimates.

Reviewer #2 (Remarks to the Author):

The authors present an impressive amount of data to show relationships between gene expression and environmental factors.

Although the language used throughout is clear, the brevity of the main paper (excluding the long “Experimental procedures” section) makes it difficult to follow at times. Clearer definitions of sets and sample sizes included in each analysis subgroup are needed throughout the paper. For example, on pages 4-5 and Table S1, does “global” refer to “all”? The methods section is currently very lengthy with a large amount of technical detail that makes it difficult to follow and to find the appropriate methods relating to each section of the results. If accepted, it is my opinion that a shortened methods section that can be clearly linked to from the results section should be included in the main paper with more technical detail (such as imputation details, etc) should be moved to the Supplement (NB: I note that the authors refer a few times in the main text to Supplementary Material and it is not clear if this means page 11 onwards in the main document or some other methods document that was missing from the reviewer download).

My major comment is about whether the authors have appropriately accounted for batch effects. The effect of “set” is explored and accounted for and the authors argue that there is no association with clinic within one of the regions (can this be quantified?). However, it is not clear whether observed

differences in gene expression between the regions (MTL, QUE and SAG) which are shown to account for ~16% of the variance of the gene expression might be the results of batch effects arising from different sample collection centres and handling procedures between the regions. In addition, it is not clear whether differences in prevalence of disease, such as asthma, between regions is driving DEG signals rather than environmental differences.

Minor comments:

Figure 1A: Does this include all 1007 individuals? Can the data in Fig 1B be presented with axes in the same orientation as Fig 1A?

Page 4: Figure 1C is referred to but should this be Fig 1D (and should Fig 1C be referred to elsewhere?).

Figure S9A: Are any of the differences across regions significant? What is the colour coding in parts B, C, D and E?

Figure S7B is unreadable as presented.

Page 7: are the references to Figures S10 and S12 the right way around?

Page 6: RDEGs are suddenly referred to but I can't find how these were defined or what this means. Please clarify or remove.

Page 6: Figure 3 is referred to but the discordant clustering of group 1 and group 2 results are not discussed. The authors state that expression profiles of DEGs and RDEGS are "largely associated" with gradients of air composition but only SO₂ demonstrates close clustering of group 1 and group 2. This should be discussed.

Figure 2: Should this be DEG (not DGE). In addition, on page 22, Fig 2 is referred to as presenting results of a coinertia analysis for DEGs and RDEGs but only DEGs are in Figure 2.

Page 10: The authors define rare variants as MAF<10%. However, rare variants are commonly defined as those that have MAF<1% (or sometimes less conservatively as those with MAF<5%). As the methods state that a 5% MAF filter was applied, the authors cannot describe results for rare variants and this needs to be re-phrased.

Page 10: is Fig S15 the correct figure?

Figure S10: how and which potential confounding factors were taken into account in this analysis?

Reviewer #3 (Remarks to the Author):

Review comments on "Gene-by-environment interactions in urban populations modulate personal risk to chronic diseases" for Nat Commun.

This extensive study aims at presenting a comprehensive analysis of environmental exposures, genetic variation and gene expression profiles. To this end, the authors used a founder population in Quebec, Canada. The authors highlight important associations between exposures and gene expression profiles, which contributes to a better understanding of gene regulation and potentially, disease mechanisms. I have the following comments on the paper.

1) In general, the paper is well-written but quite complicated to follow at a first glance. Personally, I would rather like to see the first section on regional association with expression patterns shortened somewhat, in order to leave space for the env-eQTL data which is more novel and exciting (in my view). In addition, the results section is written in a very general way with few effect estimates and measures of association (p-val, OR etc) presented, and I think this aspect can be improved to help readers navigate through the manuscript. Sometimes it is just stated that findings were replicated, but no data being presented to show how strong or consistent data actually were in the replication dataset.

2) Why look at arterial stiffness as a key determinant for initial gene expression pattern analyses? The other factors seem logic; age, sex, cell count etc but selecting only arterial stiffness from the list of available phenotypes does not make sense in my view.

3) It is stated that region of residence explains around 16% of the variance in gene expression. How much of the region effect can be explained by socio-economic factors (diet, exposures, stress etc) differing between regions? Please try to estimate this proportion.

4) I assume the NO₂ and PM_{2.5} data from LUR models used residential address history (year 2010) to obtain average individual exposure levels for the study participants? If not please clarify. Likewise, please clarify if the actual air pollution data in this study has been used in other studies or if you are only referring to the model assessment in general.

5) What type of environmental data was obtained using 3-digit postal codes? Please clarify. Have this exposure assessment been validated?

6) How were data on phenotypes obtained? Has anyone validate diagnoses and quantitative outcomes? This should be described very clearly.

7) I can't find any table listing the 34 env-eQTL genes and the interaction effects. Please add (or clarify where to find the data).

8) Page 10: "Lastly, we find evidence that suggests that personal disease risk can be modulated...." Are you really investigating disease risk here? Does the SNP-exposure-expression interaction has any direct effect on disease risk in your data? Or is this just an extrapolation from previous disease associations reported by Laprise et al? If not directly measured (please do if you can), I would recommend tuning down the disease risk statement.

9) Results from the gene-set enrichment analyses are poorly presented.

10) Why focus so much on the daily SO₂ data, and not daily variation for other exposures? I see that SO₂ showed the strongest association with phenotypes in this dataset, but to my knowledge, NO₂/NO_x and PMs have been more robustly associated with adverse health outcomes than SO₂, which makes the SO₂ focus less interesting.

11) Given the focus on air pollution effects in this study, I would recommend to also reference recent large studies on the subject, such as Ward-Caviness CK et al, Plos One 2016; Gref A et al, AJRCCM 2016 (which also includes SNP-exposure-expression analyses); Zhou Z et al, Plos One 2015 (PM_{2.5} and expression signatures in epithelial cells).

Gene-by-environment interactions in urban populations modulate personal risk to chronic disease

Response to reviewers

Our responses are in blue.

Reviewer #1 (Remarks to the Author):

This paper investigates the link between urban environment, gene expression, and genotype in a Canadian cohort. The subject is of high interest and has clear links to important public health concerns. While the results investigating differential expression between the three geographic locations is convincing and well described, I have serious concerns regarding the results presented in the latter half of the paper describing GxE interactions on gene expression. While the differential expression analysis is of interest both generally and to the transcriptomic field, the GxE analysis does not meet publication standards and needs considerable further investigation.

Major comments.

1) The inclusion of locals and internal migrants in the Region DEG analysis is nicely done and adds robustness to the conclusions that the gene expression differences due to location are more significant than due to ancestry. However, the authors should provide some caveats that there may be undetected sampling differences between the regions that could drive the changes in gene expression between the two regions. Were the samples all collected in the same year? With the same protocol at the same time of day used for collection at all locations, etc? In particular, time of year (seasonality) has a large association with gene expression in blood (Dopico, Nat Comm 2015). The authors should include time of year in their list of covariates as the sampling time in the different city seems to vary considerably– there are no collections in autumn in Seguenay for example (Figure S14). Further, air pollution in general is known to vary by time of year, and SO₂ exposure (via visual inspection of Figure S14) appears to be highest in April in all three regions.

We acknowledge that time of the year can influence gene expression levels. Participant recruitment in Quebec was between 9am to 11am in 2010 only and participant's fasting blood samples were collected between 9am to 11am. However, to control for such confounding effects on gene expression profiles, and other unwanted technical and biological variation, we used surrogate variable analysis (SVA) to generate surrogate variables that were used as covariates in the differential gene expression models. These surrogate variables correlate with technical and biological factors, including time of the year

(see Supplementary Fig. 4 - appointment date). In doing so, we control for the possible effect of sampling season on gene expression levels. Further, we added in-text details concerning the surrogate variable analysis and the technical and biological factors correlating with those variables.

In addition, we included a differential expression analysis with permutation, to reduce the possible effect of undetected sampling differences among regions (or for any two groups being compared). This is able to robustly account for outlier and batch effects. We detail our new procedures in the material and methods.

Regarding the time-point of sampling (2010), SO₂ concentrations are at the highest in April. As such, we believe it would be premature to attribute this to a seasonal effect, as our sampling timeframe was limited to one year. However, we do agree that it would be extremely interesting to study if annual changes in pollution affect gene expression over yearly cycles and we hope that our findings set a precedent for further investigation. Our sampling strategy limits us to analysis of changes across spatial and geographic areas. While, concentration and emissions of pollutants may indeed vary over the course of one year, and is certainly an interesting phenomenon with respect to blood gene expression levels, we prefer to stay cautious and restrict our interpretation to the effects of the pollutant levels themselves.

2) The authors provide a plot of the SO₂ values by region and by month of collection (Figure S14). They should add similar supplemental figures of the remaining three pollution measurements and state what QC has been done on the pollution measurements.

We added the figures for the pollutant exposure that were available at this daily scale (PM 2.5, SO₂ and O₃).

3) The authors binned the ambient pollution levels into two or three categories for the analyses. The authors should explain why they used categories rather than a continuous trait, and how they chose the cutoffs for the categories, as it appears their cutoffs led to very small sample sizes in some categories. SO₂ was categorized into two categories for DEG analysis, but 3 for GxE analysis. Why the difference?

We agree with reviewer 1 that our rationale for using categorical variables was unclear and have subsequently included details on our design in the text and the material and methods sections to provide further clarification. Further, we do not make any assumptions regarding the linearity of the relationship with the outcome (gene expression). A significant benefit of our analytical approach is that both biological interpretation and visualization are straightforward and more easily interpretable. Finally, we acknowledge that both the DGE and GxE analyses should use the same categories and we have re-run the analyses with

comparable cut off points, with two categories. For all four pollutants, and for the DGE analysis and the GXE (eQTL) analyses, we used the k-means method in R (Hartigan, J. A. and Wong, M. A. (1979). A K-means clustering algorithm. Applied Statistics 28, 100–108.) to generate the categorical variables from each of the continuous variables.

4) The set of variants used in the eQTL analysis is unclear. In the eQTL section, it is stated that SNPs were filtered to $MAF > 5\%$, and similarly in the section “Genotyping, ethnicity and regional origin of French Canadians” a MAF filter of 5% is also stated. However the text discusses discovery at rare variants, which are usually defined as $MAF < 0.1\%$. Even if the method MAF filters are mis-stated, the genotyping methods state that SNPs with $MAF < 1\%$ were filtered prior to imputation. Are all rare calls then based on imputed SNPs? This is surprising as rare variants are difficult to impute and imputed rare variants are generally considered unreliable, especially if imputed off the 1KG panel rather than a larger panel such as HRC. What info score were the imputed SNPs thresholded at? Have the imputed rare-variants been validated with sequencing or SNP typing data in this population? The authors make strong claims about the impact of rare variants on expression, these need to be supported by evidence of the accuracy of their rare variant imputation.

We thank reviewer 1 for pointing out these inconsistencies. Originally, we retained variants that had a $MAF > 5\%$ in each of the three regions independently. This led to the possibility that some variants had a $MAF > 5\%$ in one region (therefore it was retained in the analysis) but when the three regions were pooled together, a SNP may have had a $MAF < 5\%$. We refiltered our genotype and included variants that had a $MAF > 5\%$ when including all individuals from all three regions. Based on reviewer 1 comments, we have completely changed our methodology for assessing the significance of env-eQTLs in the GxE analysis (see above). We have outlined a more stringent procedure for the multiple testing. We have updated the section on rare variants by analysing the effect of these “uncommon” variants ($maf < 10\%$) and we have updated the text to match these analyses.

5) I have serious concerns regarding the significance of the GxE analysis. While the authors have used a standard method for detecting GxE on expression (matrix-eQTL cross-linear function), the testing strategy and in particular, the correction for multiple testing and the replication is not well described and appears inadequate. The GxE study design is unclear – were all genes tested for an interaction or just the ones with canonical eQTLs? It is stated that a gene-specific bonferonni correction was applied. This is unclear – were the results at each gene corrected for the number of variants tested at that gene, or the independent variants? Either way, just correcting for the number of tests at a gene is not sufficient, the authors need to correct for both the total number of genes tested as well as the number of variants. Given the high possibility of outliers driving spurious interactions (especially at rare variants), the authors should also assess whether their

interactions withstand permutations. The authors do not provide evidence of any GxE Pvalues, either uncorrected or corrected in the manuscript, these should be reported.

We thank reviewer 1 for pointing out these weaknesses. We have combined our response to comments 5 and 6 together (under (6)) as they concern the same analytic pipeline.

6) The replication of GxE interactions is unconvincing and in fact seems to show that there is no significant replication, suggesting the GxE results are likely false positives. Taking N02 as an example, the authors find 11/683 of their hits in FC are present in EURO, a replication rate of 1.6%. Can the authors assess whether this is different than the overlap expected by chance? Have the authors accounted for the number of replication tests performed in the overlap (ie what p value threshold was used in the replication set? $0.05/\text{number of signals taken into replication}$?) The authors should also confirm that the same SNP-Gene pair were tested in the discovery and replication set and that a consistent direction of effect was required between the two datasets, as these should be required for replication.

Following reviewer 1 comments, and to some extent the comments from reviewer 2 and 3, we have changed our env-eQTL analyses to incorporate the suggested multiple testing corrections and made our replication analyses more clear. The text section have also changed. Specifically:

- (1) We built discovery (n=416) and replication (n=417) cohorts from randomly selecting individuals, of any origin (FC or EURO). In this way, we avoid any differences in allele frequencies that can exist between FC and EURO that could cause differences in results. Our cohorts are roughly the same sample size and live across the spectrum of air pollution.
- (2) Our analytic pipeline is fully documented in Supplementary figure 15 and in the Material and Methods (now in the supplement). A summary is provided below:
 - (a) We perform env-eQTL modelling on the discovery cohort, and calculate p-values with a Bonferroni correction to take into account the number of eSNPs tested per gene. We then take the most significant Bonferroni-corrected p-value (i.e. eSNP-eGene pair) for each gene tested in the study (n=15632) and assess significance using an FDR threshold of 0.05.
 - (b) We perform env-eQTL modelling in the replication cohort on the significant eSNP-eGene pairs from the discovery cohort identified in (a). This set is also subject to an FDR threshold of 0.05 for assessing significance.
 - (c) For the eGenes that are significant in both the discovery and the replication cohort (n=9) (with concordant direction of effect), we perform permutation analyses to estimate “honest” p-values by resampling 1000 times 100 individuals in each exposure (for a total of n=200) and permute the individual IDs (expression levels). Thus, we are able to obtain a null distribution for the interaction effect test statistic for each eSNP-eGene pair and compare this to

the original observed test statistic obtained from the replication cohort. For each eSNP-eGene pair, we take the empirically estimated p-value to be the proportion of permutation test statistics that are larger than the observed test statistic (without permutation). We obtain several significant eSNP-eGene pairs per eGene because of linkage disequilibrium, but report only the most significant.

- (3) We have improved our discussion of the env-eQTL results by focusing on the molecular function of the significant eGenes. We have documented any membership of the eGenes to crucial gene regulatory networks, and whether any known effects of variation in sequence of expression of these eGenes on phenotypes. We have also checked whether any epigenetic marks from GM12878 cell lines (lymphoblastoid cell lines from european donor) were present within or close to any of our significant eSNP-eGene pairs that may indicate a possible effect on the eGene regulation.

7) The authors highlight a few genes linked to interesting biology, but do not state if these genes, PAX5 and AFAP1 and those highlighted in Figure S16 replicated in the two populations. The authors should clearly state if these signals replicated, and provide the p values and plots from both groups (FCs and Europeans) in figure 4 and figure S16.

We present only examples of eGenes that replicate in our discovery and replication cohorts. The material and methods, and the main text, all have been updated to reflect this.

8) The authors should include plots of all putative replicated env-eQTLs, similar to Figure S16, including the results from both datasets, pvalues and effect size estimates.

We have now included plots from all top eSNP-eGenes associations (either in Figure 4 or in Supplementary Fig. 16), together with more in-depth analysis of the eGenes: (1) whether they are part of a biologically meaningful gene network; and (2) the existence of epigenomic markers around the eGenes that could mediate environmental influences on gene expression.

Reviewer #2 (Remarks to the Author):

The authors present an impressive amount of data to show relationships between gene expression and environmental factors.

Although the language used throughout is clear, the brevity of the main paper (excluding the long “Experimental procedures” section) makes it difficult to follow at times. Clearer definitions of sets and sample sizes included in each analysis subgroup are needed throughout the paper. For example, on pages 4-5 and Table S1, does “global” refer to “all”?

The methods section is currently very lengthy with a large amount of technical detail that makes it difficult to follow and to find the appropriate methods relating to each section of the results. If accepted, it is my opinion that a shortened methods section that can be clearly linked to from the results section should be included in the main paper with more technical detail (such as imputation details, etc) should be moved to the Supplement (NB: I note that the authors refer a few times in the main text to Supplementary Material and it is not clear if this means page 11 onwards in the main document or some other methods document that was missing from the reviewer download).

We thank reviewer 2 for these comments and we have added details in the text, specifically concerning the eQTL analysis as similar concerns were raised by Reviewers One and Three. We hope that these added details will help convey our message in a more appropriate fashion.

We agree that some of our nomenclature about the sets and the replication was unclear, and we have made changes throughout the text to harmonize our definitions and to improve clarity. We changed the term 'set' with 'freeze', which is a commonly used term for different sequencing batches and will make it easier for readers to understand.

We agree that the method section is extensive, and we think that moving technical details to a supplement section would be natural. However, it is specified in the "instructions to authors" of Nature Communications to include the whole method section in the main text and not as a supplement. **We leave the decision to the editor whether this section should go in the supplement or the main text.**

My major comment is about whether the authors have appropriately accounted for batch effects. The effect of "set" is explored and accounted for and the authors argue that there is no association with clinic within one of the regions (can this be quantified?). However, it is not clear whether observed differences in gene expression between the regions (MTL, QUE and SAG) which are shown to account for ~16% of the variance of the gene expression might be the results of batch effects arising from different sample collection centres and handling procedures between the regions. In addition, it is not clear whether differences in prevalence of disease, such as asthma, between regions is driving DEG signals rather than environmental differences.

We agree that batch effect in RNAseq studies needs to be carefully taken into account. We apologize if our manuscript was unclear. Listed below are several steps that we have taken to compensate for potential batch effects. We have included a differential gene expression analysis across regions with permutations to further reduce the possible influence of latent factors, outlier individuals, or undetected sampling differences across regions (as pointed

out by reviewer one). We have added details in the manuscript and in the material and methods about these analyses.

- (1) We used surrogate variable analysis (from sva R package), that identifies and builds surrogate variables for high-dimensional data sets. SVA builds surrogate variables that are covariates constructed from high-dimensional data that can subsequently be used in downstream analyses to adjust for unknown or unwanted variation (biological or technical) and latent sources of noise (Leek and Storey 2007; Leek et al. 2012). SVA has been used extensively with various types of high-dimensional data, including RNA-sequencing ((Li, Tighe, et al. 2014; Li, Łabaj, et al. 2014; Hong et al. 2015; Gilad and Mizrahi-Man 2015)). It has been shown to be among the most effective methods for mitigating batch effects and reducing the occurrence of false positives (Li, Łabaj, et al. 2014). Indeed, we show that the surrogate variables, built from our gene expression data, correlate well with various biological and technical factors, indicating that sources of variation (biological and technical) in gene expression do exist, such as sequencing depth and cell counts. SVA corrects for these factors. In the original manuscript, we also used PEER (Stegle et al. 2012) to control for batch effects and latent variation, which has a similar approach and aim as SVA. However, because of compatibility issues with our computer cluster, we dropped PEER and relied only on SVA from now on.
- (2) To further assess for a possible remaining batch effect, or any effects driven by outlier individuals in our cohort, we also performed a differential gene expression analysis with permutations (on regions labels). First we built a null distribution of p-values for each gene based on 1000 permutations on individual's region labels. Second, we then calculated the median p-value from 1000 resampling differential expression analyses (without label permutations, and resampling 200 individuals at each iteration). We assessed significance by comparing the median p-value to the null distribution we generated with a kolmogorov-smirnov test.
- (3) We have looked into differential gene expression between clinics, but within the same region (Quebec city, Supplemental Fig.3). We do not find any differences between sampling clinics within the same region, supporting the absence of gene expression differences associated with sampling procedures between clinics.
- (4) We have also included CoIA analyses using samples of only one region (MTL - all collected at the same clinic), but that are exposed to different SO₂ levels. We replicate the patterns we see across the three regions using MTL only samples, where gene expression is associated with SO₂, and with specific phenotypes (Supplemental Fig. 12). This suggests that the environmental exposure (SO₂), and not the sampling procedures across regions, or any other variable correlated with region itself, underly these associations.

We have improved the text about these analyses. In an attempt to explore a possible effect of underlying diseases that may confound with the regional effect on gene expression, we have performed multivariate statistical analyses and we apologize if this was not clear in the original manuscript. We have improved the text to make this point clear to the readers. We used multivariate models to assess if the expression of our 170 differentially expressed genes (between high and low SO₂ exposure, that are also differentially expressed between regions) is also explained by the endophenotypes the most correlated with region. Once we regress out the effects of these phenotypes from our gene expression matrix, SO₂ still explains a significant proportion of the variance in gene expression, indicating that independent of the endophenotypes, SO₂ still explains variation in gene expression levels, across the whole province. We also performed a sensitivity with MTL only and we replicate these findings.

Minor comments:

Figure 1A: Does this include all 1007 individuals? Can the data in Fig 1B be presented with axes in the same orientation as Fig 1A?

Figure 1A and 1B represent two different PC analyses, therefore we think that comparing the PCs between them would not really benefit the readers. We have decided to show them in these orientations so they visually reflect geography better (i.e. Fig 1B: SAG is more up North, QUE in the middle, and MTL South.)

Page 4: Figure 1C is referred to but should this be Fig 1D (and should Fig 1C be referred to elsewhere?).

The text was changed according to other reviewers comments.

Figure S9A: Are any of the differences across regions significant? What is the colour coding in parts B, C, D and E?

We changed the legend accordingly.

Figure S7B is unreadable as presented.

We fixed this.

Page 7: are the references to Figures S10 and S12 the right way around?

We reviewed the figure numbering throughout the text.

Page 6: RDEGs are suddenly referred to but I can't find how these were defined or what this means. Please clarify or remove.

We clarified in the material and methods.

Page 6: Figure 3 is referred to but the discordant clustering of group 1 and group 2 results are not discussed. The authors state that expression profiles of DEGs and RDEGS are “largely associated” with gradients of air composition but only SO₂ demonstrates close clustering of group 1 and group 2. This should be discussed.

We have added this observation to our discussion and rational for choosing SO₂ as the variable to explore more deeply.

Figure 2: Should this be DEG (not DGE). In addition, on page 22, Fig 2 is referred to as presenting results of a coinertia analysis for DEGs and RDEGs but only DEGs are in Figure 2.

Thank you for pointing out this writing mistake.

Page 10: The authors define rare variants as MAF<10%. However, rare variants are commonly defined as those that have MAF<1% (or sometimes less conservatively as those with MAF<5%). As the methods state that a 5% MAF filter was applied, the authors cannot describe results for rare variants and this needs to be re-phrased.

We have changed our nomenclature and now use “uncommon” variants to refer to those variants with MAF < 10%. We have also changed our filtering for MAF following other’s reviewers comments.

Page 10: is Fig S15 the correct figure?

This text part was changed following other reviewer’s comments.

Figure S10: how and which potential confounding factors were taken into account in this analysis?

This analysis shows the correlations between endophenotypes and environmental factors. Some are indeed highly correlated (i.e. several cardiovascular phenotypes are correlated, such as peripheral AIX, central AIX, and arterial stiffness), and this figure allow the reader to appreciate these correlations, or the absence of such correlations (i.e. Age is not correlated with any of the environmental variables: this indicates that participants of the same age do not cluster within a particular environmental exposure) .

Reviewer #3 (Remarks to the Author):

Review comments on “Gene-by-environment interactions in urban populations modulate personal risk to chronic diseases” for Nat Commun.

This extensive study aims at presenting a comprehensive analysis of environmental exposures, genetic variation and gene expression profiles. To this end, the authors used a founder population in Quebec, Canada. The authors highlight important associations

between exposures and gene expression profiles, which contributes to a better understanding of gene regulation and potentially, disease mechanisms. I have the following comments on the paper.

1) In general, the paper is well-written but quite complicated to follow at a first glance. Personally, I would rather like to see the first section on regional association with expression patterns shortened somewhat, in order to leave space for the env-eQTL data which is more novel and exciting (in my view). In addition, the results section is written in a very general way with few effect estimates and measures of association (p-val, OR etc) presented, and I think this aspect can be improved to help readers navigate through the manuscript. Sometimes it is just stated that findings were replicated, but no data being presented to show how strong or consistent data actually were in the replication dataset.

These comments are consistent with concerns raised by other reviewers and we have reduced the length of the regional association patterns, and have improved our discussion of the env-eQTLs. Further, we have added a more in-depth discussion of the env-eQTL loci, including gene function, networks, and the presence of epigenetic marks in our significant env-eQTLs.

2) Why look at arterial stiffness as a key determinant for initial gene expression pattern analyses? The other factors seem logic; age, sex, cell count etc but selecting only arterial stiffness from the list of available phenotypes does not make sense in my view.

While this error was mentioned in the text, it was not used in the analyses that were presented (see Figure 1e and Supplemental Fig. 2). Only age, sex, cell counts, gender and region were used. We have removed it from the text (this section was partially moved to the material and methods following reviewer's Three comments about the first section on gene expression being too lengthy).

3) It is stated that region of residence explains around 16% of the variance in gene expression. How much of the region effect can be explained by socio-economic factors (diet, exposures, stress etc) differing between regions? Please try to estimate this proportion.

We have improved the section where we test for the contribution of all environmental variables on the regional effect on gene expression by including multivariate models to test whether and how much the measured environmental variables contribute to the regional effect (gene expression differences in the top 170 differentially expressed genes). We find that the regional effect on the gene expression is mostly associated with ambient air pollution, and less so, or not at all, with diseases, smoking, or socio-economic factors that were measured. All results are presented in Supplementary Table 7.

4) I assume the NO₂ and PM_{2.5} data from LUR models used residential address history (year 2010) to obtain average individual exposure levels for the study participants? If not please clarify. Likewise, please clarify if the actual air pollution data in this study has been used in other studies or if you are only referring to the model assessment in general.

See point (5) for a complete answer to both point (4) and (5) as they relate to the same data.

5) What type of environmental data was obtained using 3-digit postal codes? Please clarify. Have this exposure assessment been validated?

All environmental data described in the study was obtained using the 3-digit postal codes of participants. We have clarified our main text to eliminate ambiguity. The clarified text reads as follows: “A total of 12 environmental exposures were included, all of them measured or estimated at the level of three-digit postal code”.

The environmental data has been made available to the public (e.g. NAPS data, satellite-LUR PM_{2.5}, built environment and socio-economic indicators), either through open-source databases from the Government of Canada and/or Government of Québec, or scientific publications. It has been extensively used for health risk estimation with very large cohorts (e.g., Crouse et al. 2015 <https://ehp.niehs.nih.gov/14-09276/>) and for the global burden of disease calculation (e.g., Brauer et al., 2016; Forouzanfar et al., 2016; van Donkelaar et al., 2015; Lim et al., 2012). To our knowledge, this is the best and most reliable available environmental exposure data available across Canada. We have added details on Supplementary Table 4 to indicate where each of the environmental data was taken from.

6) How were data on phenotypes obtained? Has anyone validate diagnoses and quantitative outcomes? This should be described very clearly.

These comments have been taken into consideration. All phenotypes were obtained through CARTaGENE health tests or health questionnaires. All self-reported diagnoses for diseases were cross-validated with electronic health records from the universal health care system of Quebec province (RAMQ). The whole detailed protocol is available in the work of Awadalla et al. 2013 that we cite, and we have added details in the text to highlight these validations.

7) I can't find any table listing the 34 env-eQTL genes and the interaction effects. Please add (or clarify where to find the data).

We added a table (Supplementary Table 9) describing our most significant env-eQTLs that survive multiple testing and replication.

8) Page 10: “Lastly, we find evidence that suggests that personal disease risk can be modulated.....” Are you really investigating disease risk here? Does the SNP-exposure-expression interaction has any direct effect on disease risk in your data? Or is this just an extrapolation from previous disease associations reported by Laprise et al? If not directly measured (please do if you can), I would recommend tuning down the disease risk statement.

We thank reviewer Three for this comment. Our env-eQTL top hits have changed with the new pipeline, and we therefore do not discuss the findings of Laprise et al. as they are not relevant to our top env-eQTLs anymore. We have taken this comment into consideration and turned down the disease risk statement.

9) Results from the gene-set enrichment analyses are poorly presented.

We thank reviewer 3 for the comment, however, it is unclear to us which figure he/she is referring to. We have removed part b of Supplementary Fig. 7, as it is redundant with Supplementary Table 3. All other graphs showing GO term enrichment summaries used standard ways of graphing results for GO terms summaries. We are open to any detailed suggestions about how to present those results differently.

10) Why focus so much on the daily SO₂ data, and not daily variation for other exposures? I see that SO₂ showed the strongest association with phenotypes in this dataset, but to my knowledge, NO₂/NO_x and PMs have been more robustly associated with adverse health outcomes than SO₂, which makes the SO₂ focus less interesting.

In our cohort, we noted that the high levels of SO₂ exposure are significantly associated with detrimental effects on cardio-respiratory phenotypes, more so than annual PM_{2.5} and annual NO₂ ambient levels, and SO₂ effect replicate better across groups (Fig. 3). We do not imply by any means that PM_{2.5} and NO₂/NO_x are not detrimental, but in our system, the relative effect of these pollutants is less than SO₂. As O₃ levels are more dependable on other various ambient factors (sunlight, other NO_x emissions), we decided to focus our high-resolution analyses on the participant’s weekly SO₂ exposure. We thank reviewer 3 for pointing this out and have now added a few sentences in our manuscript to help the readers understand the rationale behind it.

11) Given the focus on air pollution effects in this study, I would recommend to also reference recent large studies on the subject, such as Ward-Caviness CK et al, Plos One

2016; Gref A et al, AJRCCM 2016 (which also includes SNP-exposure-expression analyses); Zhou Z et al, Plos One 2015 (PM2.5 and expression signatures in epithelial cells).

We thank reviewer Three for these important suggestions of large recent studies and we have added them as references.

References

- Gilad, Yoav, and Orna Mizrahi-Man. 2015. "A Reanalysis of Mouse ENCODE Comparative Gene Expression Data." *F1000Research* 4 (May). ncbi.nlm.nih.gov: 121.
- Hong, Xiumei, Ke Hao, Christine Ladd-Acosta, Kasper D. Hansen, Hui-Ju Tsai, Xin Liu, Xin Xu, et al. 2015. "Genome-Wide Association Study Identifies Peanut Allergy-Specific Loci and Evidence of Epigenetic Mediation in US Children." *Nature Communications* 6 (February). nature.com: 6304.
- Leek, Jeffrey T., W. Evan Johnson, Hilary S. Parker, Andrew E. Jaffe, and John D. Storey. 2012. "The Sva Package for Removing Batch Effects and Other Unwanted Variation in High-Throughput Experiments." *Bioinformatics* 28 (6). academic.oup.com: 882–83.
- Leek, Jeffrey T., and John D. Storey. 2007. "Capturing Heterogeneity in Gene Expression Studies by Surrogate Variable Analysis." *PLoS Genetics* 3 (9): 1724–35.
- Li, Sheng, Paweł P. Łabaj, Paul Zumbo, Peter Sykacek, Wei Shi, Leming Shi, John Phan, et al. 2014. "Detecting and Correcting Systematic Variation in Large-Scale RNA Sequencing Data." *Nature Biotechnology* 32 (9): 888–95.
- Li, Sheng, Scott W. Tighe, Charles M. Nicolet, Deborah Grove, Shawn Levy, William Farmerie, Agnes Viale, et al. 2014. "Multi-Platform Assessment of Transcriptome Profiling Using RNA-Seq in the ABRF next-Generation Sequencing Study." *Nature Biotechnology* 32 (9). nature.com: 915–25.
- Stegle, Oliver, Leopold Parts, Matias Piipari, John Winn, and Richard Durbin. 2012. "Using Probabilistic Estimation of Expression Residuals (PEER) to Obtain Increased Power and Interpretability of Gene Expression Analyses." *Nature Protocols* 7 (3): 500–507.

Reviewers' comments:

Reviewer #1 (Remarks to the Author, please also see the pdf file):

Gene-by-environment interactions in urban populations
modulate personal risk to chronic disease

Response to reviewers

Our responses are in blue .

Reviewer 1 response to rebuttal in red (have uploaded annotated file as attachment so colors can be seen)

Reviewer #1 (Remarks to the Author):

This paper investigates the link between urban environment, gene expression, and genotype in a Canadian cohort. The subject is of high interest and has clear links to important public health concerns. While the results investigating differential expression between the three geographic locations is convincing and well described, I have serious concerns regarding the results presented in the latter half of the paper describing GxE interactions on gene expression. While the differential expression analysis is of interest both generally and to the transcriptomic field, the GxE analysis does not meet publication standards and needs considerable further investigation.

Major comments.

1) The inclusion of locals and internal migrants in the Region DEG analysis is nicely done and adds robustness to the conclusions that the gene expression differences due to location are more significant than due to ancestry. However, the authors should provide some caveats that there may be undetected sampling differences between the regions that could drive the changes in gene expression between the two regions. Were the samples all collected in the same year? With the same protocol at the same time of day used for collection at all locations, etc? In particular, time of year (seasonality) has a large association with gene expression in blood (Dopico, Nat Comm 2015). The authors should include time of year in their list of covariates as the sampling time in the different city seems to vary considerably– there are no collections in autumn in Seguenay for example (Figure S14). Further, air pollution in general is known to vary by time of year, and SO₂ exposure (via visual inspection of Figure S14) appears to be highest in April in all three regions.

We acknowledge that time of the year can influence gene expression levels. Participant recruitment in Quebec was between 9am to 11am in 2010 only and participant's fasting blood samples were collected between 9am to 11am. However, to control for such confounding effects on gene expression profiles, and other unwanted technical and biological variation, we used surrogate variable analysis (SVA) to generate surrogate variables that were used as covariates in the differential gene expression models. These surrogate variables correlate with technical and biological factors, including time of the year (see Supplementary Fig. 4 - appointment date). In doing so, we control for the possible effect of sampling season on gene expression levels. Further, we added in-text details concerning the surrogate variable analysis and the technical and biological factors correlating with those variables.

In addition, we included a differential expression analysis with permutation, to reduce the possible effect of undetected sampling differences among regions (or for any two groups being compared). This is able to robustly account for outlier and batch effects. We detail our new procedures in the material and methods.

Regarding the time-point of sampling (2010), SO₂ concentrations are at the highest in April. As such, we believe it would be premature to attribute this to a seasonal effect, as our sampling timeframe was limited to one year. However, we do agree that it would be extremely interesting to study if annual changes in pollution affect gene expression over yearly cycles and we hope that our findings set a

precedent for further investigation. Our sampling strategy limits us to analysis of changes across spatial and geographic areas. While, concentration and emissions of pollutants may indeed vary over the course of one year, and is certainly an interesting phenomenon with respect to blood gene expression levels, we prefer to stay cautious and restrict our interpretation to the effects of the pollutant levels themselves.

My primary concern is that seasonal effects on gene expression could be mistaken for pollution-expression effects if season and pollution are highly correlated, which is well established externally and shown very clearly in their SO₂ and PM_{2.5} data. I was not suggesting that the authors should attempt to identify seasonal effects on pollution in their one year window – this is well established externally. The authors point out that the SVA's account for appointment date, which should account for seasonal effects in SO₂ and O₃ based on suppl Figure 14, as such I consider this matter resolved from an analytical standpoint, but do think that a caveat should be added to the text about the potential confounding, especially for PM_{2.5} where appointment date is not linearly correlated with the pollutant. The authors state they added in text details, but did not provide line numbers or tracked changes in the document, so perhaps this has already been added and I missed it, and when I searched the text for the word 'season' no hits were found.

2) The authors provide a plot of the SO₂ values by region and by month of collection (Figure S14). They should add similar supplemental figures of the remaining three pollution measurements and state what QC has been done on the pollution measurements.

We added the figures for the pollutant exposure that were available at this daily scale (PM 2.5, SO₂ and O₃).

RESOLVED

3) The authors binned the ambient pollution levels into two or three categories for the analyses. The authors should explain why they used categories rather than a continuous trait, and how they chose the cutoffs for the categories, as it appears their cutoffs led to very small sample sizes in some categories. SO₂ was categorized into two categories for DEG analysis, but 3 for GxE analysis. Why the difference?

We agree with reviewer 1 that our rationale for using categorical variables was unclear and have subsequently included details on our design in the text and the material and methods sections to provide further clarification. Further, we do not make any assumptions regarding the linearity of the relationship with the outcome (gene expression). A significant benefit of our analytical approach is that both biological interpretation and visualization are straightforward and more easily interpretable. Finally, we acknowledge that both the DGE and GxE analyses should use the same categories and we have re-run the analyses with comparable cut off points, with two categories. For all four pollutants, and for the DGE analysis and the GXE (eQTL) analyses, we used the k-means method in R (Hartigan, J. A. and Wong, M. A. (1979). A K-means clustering algorithm. Applied Statistics 28, 100–108.) to generate the categorical variables from each of the continuous variables.

I thank the authors for re-running the analysis with the same categories in both DGE and GxE and I think the k-means method described here to generate the categorical variables. However I could find no text in the document describing how they selected the new categories(including the citation above), the actual cutoff used or the number of samples in each category. These should all be added. It is difficult to interpret the results without knowing the sample size in the different categories.

4) The set of variants used in the eQTL analysis is unclear. In the eQTL section, it is stated that SNPs were filtered to MAF > 5%, and similarly in the section "Genotyping, ethnicity and regional origin of French Canadians" a MAF filter of 5% is also stated. However the text discusses discovery at rare variants, which are usually defined as MAF < 0.1%. Even if the method MAF filters are mis-stated,

the genotyping methods state that SNPs with $MAF < 1\%$ were filtered prior to imputation. Are all rare calls then based on imputed SNPs? This is surprising as rare variants are difficult to impute and imputed rare variants are generally considered unreliable, especially if imputed off the 1KG panel rather than a larger panel

such as HRC. What info score were the imputed SNPs thresholded at? Have the imputed rare-variants been validated with sequencing or SNP typing data in this population? The authors make strong claims about the impact of rare variants on expression, these need to be supported by evidence of the accuracy of their rare variant imputation.

We thank reviewer 1 for pointing out these inconsistencies. Originally, we retained variants that had a $MAF > 5\%$ in each of the three regions independently. This led to the possibility that some variants had a $MAF > 5\%$ in one region (therefore it was retained in the analysis) but when the three regions were pooled together, a SNP may have had a $MAF < 5\%$. We refiltered our genotype and included variants that had a $MAF > 5\%$ when including all individuals from all three regions. Based on reviewer 1 comments, we have completely changed our methodology for assessing the significance of env-eQTLs in the GxE analysis (see above). We have outlined a more stringent procedure for the multiple testing. We have updated the section on rare variants by analysing the effect of these "uncommon" variants ($maf < 10\%$) and we have updated the text to match these analyses.

I appreciate the authors efforts to provide a consistent genotype filtering scheme, however I have serious issues with the 'uncommon' variant narrative. The firmly established field standard is to define variants with $MAF > 5\%$ as common. The authors do not provide the MAF of their lead SNPs in the text or tables. I put all 7 lead SNPs from supplemental table 9 into ensembl to calculate the MAF in the 1000 genomes European populations. I took the value from "EUR" – rather than individual populations, but nearly all sub-populations in 1000 genomes were identical to EUR, so this should be a good proxy for the French Canadian MAF. Of the 7 lead SNPs, one had MAF 37%, five had MAF 9-10% and one had MAF of 5%. It is extremely unorthodox to call variants of MAF 9% 'uncommon' and the highly-experienced authors of this manuscript should be well aware of this. Both the original and revised text reads as if the authors are intent on forcing their results into a pre-selected rare/uncommon narrative that the results do not support

Supplementary Figure 17 purports to show that less common variants have a larger effect size. However, the peak effect size appears to be at the MAF of the lead SNP for each gene (5/7 of which are MAF of $\sim 10\%$). The fall off in effect size at eSNPs in the region is a result of residual – LD between the causal variant (tagged by the lead SNP) and the other SNPs in the region. To make claims about the relative effect sizes of variants of different MAF the authors should compare the effect sizes of the lead SNP between different genes, not between all SNPs at a given locus that are in variable LD with the lead SNP.

The analysis presented in the section "uncommon genetic variants and environmental factors synergistically modulate phenotypic variation" is opaque and there is no methods section for this analysis. I cannot determine what analysis was done. The figure cited for these results, Supplementary Figure 18 is also unclear. What SNPs were included in this enrichment? The seven lead SNPs from the GxE? The figure legend says table 7, which does not exist, implies that perhaps all SNPs in the region? Are the authors attempting to define if these 7 SNPs are enriched/coincident for association to phenotypes? If so, it would be far more appropriate to look them up in large GWAS's, which are available for nearly all of these traits, rather than test association in 400 individuals

5) I have serious concerns regarding the significance of the GxE analysis. While the authors have used a standard method for detecting GxE on expression (matrix-eQTL cross-linear function), the testing strategy and in particular, the correction for multiple testing and the replication is not well described

and appears inadequate. The GxE study design is unclear – were all genes tested for an interaction or just the ones with canonical eQTLs? It is stated that a gene-specific bonferonni correction was applied. This is unclear – were the results at each gene corrected for the number of variants tested at that gene, or the independent variants? Either way, just correcting for the number of tests at a gene is not sufficient, the authors need to correct for both the total number of genes tested as well as the number of

variants. Given the high possibility of outliers driving spurious interactions (especially at rare variants), the authors should also assess whether their

interactions withstand permutations. The authors do not provide evidence of any GxE Pvalues, either uncorrected or corrected in the e manuscript, these should be reported.

We thank reviewer 1 for pointing out these weaknesses. We have combined our response to comments 5 and 6 together (under (6)) as they concern the same analytic pipeline.

6) The replication of GxE interactions is unconvincing and in fact seems to show that there is no significant replication, suggesting the GxE results are likely false positives. Taking N02 as an example, the authors find 11/683 of their hits in FC are present in EURO, a replication rate of 1.6%. Can the authors assess whether this is different than the overlap expected by chance? Have the authors accounted for the number of replication tests performed in the overlap (ie what p value threshold was used in the replication set? 0.05/number of signals taken into replication?) The authors should also confirm that the same SNP-Gene pair were tested in the discovery and replication set and that a consistent direction of effect was required between the two datasets, as these should be required for replication.

Following reviewer 1 comments, and to some extent the comments from reviewer 2 and 3, we have changed our env-eQTL analyses to incorporate the suggested multiple testing corrections and made our replication analyses more clear. The text section have also changed. Specifically: (1) We built discovery (n=416) and replication (n=417) cohorts from randomly selecting individuals, of any origin (FC or EURO). In this way, we avoid any differences in allele frequencies that can exist between FC and EURO that could cause differences in results. Our cohorts are roughly the same sample size and live across the spectrum of air pollution. (2) Our analytic pipeline is fully documented in Supplementary figure 15 and in the Material and Methods (now in the supplement). A summary is provided below: (a) We perform env-eQTL modelling on the discovery cohort, and calculate p-values with a Bonferroni correction to take into account the number of eSNPs tested per gene. We then take the most significant Bonferroni-corrected p-value (i.e. eSNP-eGene pair) for each gene tested in the study (n=15632) and assess significance using an FDR threshold of 0.05. (b) We perform env-eQTL modelling in the replication cohort on the significant eSNP-eGene pairs from the discovery cohort identified in (a). This set is also subject to an FDR threshold of 0.05 for assessing significance. (c) For the eGenes that are significant in both the discovery and the replication cohort (n=9) (with concordant direction of effect), we perform permutation analyses to estimate “honest” p-values by resampling 1000 times 100 individuals in each exposure (for a total of n=200) and permute the individual IDs (expression levels). Thus, we are able to obtain a null distribution for the interaction effect test statistic for each eSNP-eGene pair and compare this to the original observed test statistic obtained from the replication cohort. For each eSNP-eGene pair, we take the empirically estimated p-value to be the proportion of permutation test statistics that are larger than the observed test statistic (without permutation). We obtain several significant eSNP-eGene pairs per eGene because of linkage disequilibrium, but report only the most significant. (3) We have improved our discussion of the env-eQTL results by focusing on the molecular function of the significant eGenes. We have documented any membership of the eGenes to crucial gene regulatory networks, and whether any known effects of variation in sequence of expression of these eGenes on phenotypes. We have also checked whether any epigenetic marks from GM12878 cell lines (lymphoblastoid cell lines from european donor) were present within or close to any of our significant eSNP-eGene pairs that may indicate a possible effect on the eGene regulation.

I appreciate the efforts the authors have made to modify their analytical pipeline, which are certainly non-trivial, however I still have major concerns with this analysis. Many key details of the analysis are missing which make the end results impossible to interpret. Information needed includes:

- How many eSNP-eGene pairs were called significant in the discovery set and taken into replication?
- How was the FDR in the replication dataset determined?
- Why does the permutation re-sampling only include 200 re-sampled IDS rather than permuting across all IDs? The test statistics from the permutation are then compared “to the original observed test statistic obtained from the replication cohort” However as the replication cohort has an N of 400 this is a completely invalid comparison – the larger sample size is expected to have more significant test statistic. The ‘honest’ permutation pvalue is uninterpretable.
- For the permutation: “We obtain several significant eSNP-eGene pairs per eGene because of linkage disequilibrium, but report only the most significant” How is this possible if only the lead SNP-Genes Pair was taken into replication? Is the permutation testing a different set of SNPs than the one identified in discovery and tested in the replication dataset? If so this is not a valid approach.
- Sup Table 9 reports both empirical permutation p-values (only 4/7 of which are less than 0.05) and an asymptotic pvalue, which are more significant. First, the text should clearly state that three of the reported genes failed the permutation test (although as noted above, the calculation of the empirical p value needs to be modified). There is no mention of the asymptotic. P value in text or methods, this should be removed or explained.
- The authors provide no information on the results from the replication and discovery for their hits – they should add the Pvalue, effect size, standard error estimates from all 3 stages (discovery, replication and combined analysis) to Sup Table 9, along with the MAF and effect allele of the SNPs, a subset of this information is reported in the text for two highlighted genes, but should be provided for all genes in one table.
- The legend of Table S9 says that it shows “significant top eSNPs-eGene pairs of env-eQTLs after permutation tests (n = 1000) from two cohorts, combined” This strongly suggests that the permutation was done on the full dataset, N =800, and should be reworded.

7) The authors highlight a few genes linked to interesting biology, but do not state if these genes, PAX5 and AFAP1 and those highlighted in Figure S16 replicated in the two populations. The authors should clearly state if these signals replicated, and provide the p values and plots from both groups (FCs and Europeans) in figure 4 and figure S16.

We present only examples of eGenes that replicate in our discovery and replication cohorts. The material and methods, and the main text, all have been updated to reflect this.

RESOLVED

8) The authors should include plots of all putative replicated env-eQTLs, similar to FigureS16, including the results from both datasets, pvalues and effect size estimates.

We have now included plots from all top eSNP-eGenes associations (either in Figure 4 or in Supplementary Fig. 16), together with more in-depth analysis of the eGenes: (1) whether they are part of a biologically meaningful gene network; and (2) the existence of epigenomic markers around the eGenes that could mediate environmental influences on gene expression.

I appreciate the additional Supplemental plots. The authors should state whether the plots are from the discovery dataset or the combined analysis. As noted above, the authors should also add the pvalues and effect size estimates from discovery, replication and combined analysis to Sup Table 9,

along with the MAF and effect allele of the SNP, which have not been added despite previous request above. I acknowledge the discovery p values are listed for some genes in the text, but this should be comprehensively listed for all genes.

Reviewer #2 (Remarks to the Author):

I am happy that my comments have been addressed.

Reviewer #3 (Remarks to the Author):

The authors have adequately addressed my comments. No further comments from my end.

Gene-by-environment interactions in urban populations modulate personal risk to chronic diseases

Reviewer 1 **round 2** in red

Our responses in blue

(1) My primary concern is that seasonal effects on gene expression could be mistaken for pollution-expression effects if season and pollution are highly correlated, which is well established externally and shown very clearly in their SO₂ and PM_{2.5} data. I was not suggesting that the authors should attempt to identify seasonal effects on pollution in their one year window – this is well established externally. The authors point out that the SVA's account for appointment date, which should account for seasonal effects in SO₂ and O₃ based on suppl Figure 14, as such I consider this matter resolved from an analytical standpoint, but do think that a caveat should be added to the text about the potential confounding, especially for PM_{2.5} where appointment date is not linearly correlated with the pollutant. The authors state they added in text details, but did not provide line numbers or tracked changes in the document, so perhaps this has already been added and I missed it, and when I searched the text for the word 'season' no hits were found.

We thank the reviewer for acknowledging that we have resolved the issue analytically. We added in the text a sentence about the potential confounding factor between season and pollution on gene expression : “It is widely known that ambient air pollution covaries with season, and we accounted for blood collection date in our models (Supplementary Fig. 4c). However, we cannot fully exclude a possible residual contribution of season on gene expression patterns.”

(2) Resolved

(3) I thank the authors for re-running the analysis with the same categories in both DGE and GxE and I think the k-means method described here to generate the categorical variables. However I could find no text in the document describing how they selected the new categories(including the citation above), the actual cutoff used or the number of samples in each category. These should all be added. It is difficult to interpret the results without knowing the sample size in the different categories

We included details and citation of the kmeans clustering and we added the cluster means and number of individuals in each category in the method section:

“We categorized the participants using a k-means algorithm (Hartigan and Wong 1979) into two categories high exposure or low exposure categories, irrespective of the pollutant type. A k-means algorithm attempts to partition the data points into k groups (here, $k=2$), such that the sum of squared Euclidean distances squares from points to the assigned cluster centroidcentres (cluster mean) is minimized.”

Table 1: Summary of k-mean clustering. Cluster means and number of individuals within each categories.

	Low exposure		High exposure	
Pollutant	Cluster mean by pollutants	Number of individuals	cluster mean by pollutants	Number of individuals
PM2.5	8.95	392	5.97	605
NO2	5.86	160	14.34	837
O3	22.97	775	25.05	222
SO2	0.72	339	1.90	658

(4) I appreciate the authors efforts to provide a consistent genotype filtering scheme, however I have serious issues with the ‘uncommon’ variant narrative. The firmly established field standard is to define variants with MAF > 5% as common. The authors do not provide the MAF of their lead SNPs in the text or tables. I put all 7 lead SNPs from supplemental table 9 into ensembl to calculate the MAF in the 1000 genomes European populations. I took the value from “EUR” – rather than individual populations, but nearly all sub-populations in 1000 genomes were identical to EUR, so this should be a good proxy for the French Canadian MAF. Of the 7 lead SNPs, one had MAF 37%, five had MAF 9-10% and one had MAF of 5%. It is extremely unorthodox to call variants of MAF 9% ‘uncommon’ and the highly-experienced authors of this manuscript should be well aware of this. Both the original and revised text reads as if the authors are intent on forcing their results into a pre-selected rare/uncommon narrative that the results do not support

We apologize for the confusion regarding the analyses presented. We never meant to consider all of our lead SNP as “uncommon”, rather less common variants, and we

apologize if our text and methodology were misleading. We think that this confusion arose from our wording in the section “Uncommon genetic variants and environmental factors synergistically modulate phenotypic variation”. **Our intention was to show that env-eQTL effect sizes are inversely correlated with allele MAF.** We revised the text by including more details and changing the narrative, and improved the methods, and the legend of Supplementary figure 18 to clarify the methods of our analyses and results. We added MAF in our cohort of lead SNPs in Supplementary Table 9. We have also included Reviewer’s 1 suggestion of looking at the relationship between MAF and effect size (Supplementary fig 17), but for lead SNPs only. We thank reviewer 1 for this suggestion as this made our results more easily interpretable for the readers. We looked up the 7 lead SNPs in the GWAS catalogue but did not find any association with traits.

(5) and (6) I appreciate the efforts the authors have made to modify their analytical pipeline, which are certainly non-trivial, however I still have major concerns with this analysis. Many key details of the analysis are missing which make the end results impossible to interpret. Information needed includes:

- How many eSNP-eGene pairs were called significant in the discovery set and taken into replication?

We have 10 hits of lead eSNPs retained from the discovery, and 7 survive replication (Supplementary table 9). From these 7 pairs, 5 also show significance from the combined cohort analysis, as well as from empirical p-values calculated from permutations, and have consistent direction of effect size across all of the cohorts (bold entries in Supplementary table 9). This is clarified in the main text section “Environmental factors modulate the penetrance of genetic variants”.

- How was the FDR in the replication dataset determined?

FDR was determined by obtaining q-values for the 10 SNPs tested in the replication cohort. We have added clarification to the relevant section of the main text and the methods.

- Why does the permutation re-sampling only include 200 re-sampled IDS rather than permuting across all IDs? The test statistics from the permutation are then compared “to the original observed test statistic obtained from the replication cohort” However as the replication cohort has an N of 400 this is a completely invalid comparison – the larger

sample size is expected to have more significant test statistic. The 'honest' permutation pvalue is uninterpretable.

We thank reviewer 1 and apologize for this mistake. We have realized that an incorrect version of the methods text has been used, and was not reflecting our current analyses. Indeed we did **not** resample a subset of 200 individuals, but in fact the whole cohort, as depicted in Supplementary fig 15 and mentioned originally in Supplementary table 9. We have revised the text to describe the methods correctly.

- For the permutation: "We obtain several significant eSNP-eGene pairs per eGene because of linkage disequilibrium, but report only the most significant" How is this possible if only the lead SNP-Gene Pair was taken into replication? Is the permutation testing a different set of SNPs than the one identified in discovery and tested in the replication dataset? If so this is not a valid approach.

We apologize for creating unnecessary confusion. Our comment regarding LD in the text is inaccurate with respect to our current methodology (where only the lead SNP is considered a candidate) and we have removed it. We do obtain 10 significant hits in the discovery, but indeed after the replication only 7 associations remain as significant (reported in Supplementary table 9). We have revised this text accordingly.

- Sup Table 9 reports both empirical permutation p-values (only 4/7 of which are less than 0.05) and an asymptotic pvalue, which are more significant. First, the text should clearly state that three of the reported genes failed the permutation test (although as noted above, the calculation of the empirical p value needs to be modified). There is no mention of the asymptotic. P value in text or methods, this should be removed or explained.

We have removed the word asymptotic as it may lead to confusion. We have also revised the empirical p-values by correcting a small mistake in the calculation, hence the slightly different empirical p-values in the latest Supplementary Table 9.

- The authors provide no information on the results from the replication and discovery for their hits – they should add the Pvalue, effect size, standard error estimates from all 3 stages (discovery, replication and combined analysis) to Sup Table 9, along with the MAF and effect allele of the SNPs, a subset of this information is reported in the text for two highlighted genes, but should be provided for all genes in one table.

We agree that this information is needed and we have updated our tables and legends.

- The legend of Table S9 says that it shows "significant top eSNPs-eGene pairs of env-eQTLs after permutation tests (n = 1000) from two cohorts, combined" This strongly

suggests that the permutation was done on the full dataset, N =800, and should be reworded.

Indeed, permutation was done on the full dataset. As mentioned previously, we have updated our methods and legends.

(7) Resolved

(8) I appreciate the additional Supplemental plots. The authors should state whether the plots are from the discovery dataset or the combined analysis. As noted above, the authors should also add the p-values and effect size estimates from discovery, replication and combined analysis to Sup Table 9, along with the MAF and effect allele of the SNP, which have not been added despite previous request above. I acknowledge the discovery p values are listed for some genes in the text, but this should be comprehensively listed for all genes.

The plots are from the combined cohort, and we have added that information to the figure. We apologize for missing your previous request and have subsequently added these details to Supplementary Table 9.

Reviewers' comments:

Reviewer #1 (Remarks to the Author):

1) Resolved

2) Previously resolved

3) Resolved

4) Resolved – but to avoid confusion with the commonly accepted terms used in the field (where > 0.05 is considered 'common') the authors should clearly state in their text and figures that their "less common" variant category are in fact MAF between 0.05 and 0.1, not merely that it is < 0.1 .

5-6) I appreciate the additional information added to the manuscript in the new tables and additions to tables as well as the correction to the methods. However, I am still confused on two points:

A) There is an inconsistency in the reporting of the env eQTL results. The methods section states (appropriately) that a consistent direction of effect was required for replication. "We then reported the environmental eSNP-eGene pairs that were significant (replicated) in the replication cohort ($FDR < 0.05$) and had the same direction of effect in both cohorts." However three of the seven pairs reported in sup table 19 and discussed in the text have a different direction of effect in discovery and replication in table 19, and the table legend even states that: "Two genes with small effect sizes show discordance of effect size in either the replication or the combined datasets". Opposite directions of effect is a strong indication of false positives and should not be referred to as a replication.

 The authors should revise the text, figures and abstract to clearly state that four out of 10 SNP-eGene pairs taken into replication replicated, not seven. **

B) I am still confused on how FDR in the replication cohort was calculated. Please explicitly state how the FDR in the replication set was calculated – Sup Figure 15 states "FDR (BH at 0.05) on nb of total genes". What does nb mean? Does total eGenes here mean all quantified or just those significant in discovery? The authors rebuttal stated "FDR was determined by obtaining q-values for the 10 SNPs tested in the replication cohort. We have added clarification to the relevant section of the main text and the methods." How do you obtain a q-value for only 10 SNPs? No details have been added to the methods to clarify how FDR in the replication cohort was determined.

7-8) resolved

Finally, given the revised GxE results and the removal of any results directly linking genetic variants to chronic disease in the manuscript, the title of the manuscript "Gene-by-environment interactions in urban populations modulate personal risk to chronic disease" should be modified. This is consistent with the previous request from Reviewer 3 to "If not directly measured (please do if you can), I would recommend tuning down the disease risk statement." And the authors reply to this request "We have taken this comment into consideration and turned down the disease risk statement." The authors did indeed tone down the text, but the title still reflects the original submitted manuscript rather than the revised version.

Reviewers' comments in red

Our response in blue

Reviewer #1 (Remarks to the Author):

1) Resolved

2) Previously resolved

3) Resolved

4) Resolved – but to avoid confusion with the commonly accepted terms used in the field (where > 0.05 is considered 'common') the authors should clearly state in their text and figures that their “less common” variant category are in fact MAF between 0.05 and 0.1, not merely that it is < 0.1 .

We modified the main text, methods, and Supplemental figure 18 that our less common variants category includes variants of MAF between 0.05 and 0.1

5-6) I appreciate the additional information added to the manuscript in the new tables and additions to tables as well as the correction to the methods. However, I am still confused on two points:

A) There is an inconsistency in the reporting of the env eQTL results. The methods section states (appropriately) that a consistent direction of effect was required for replication. “We then reported the environmental eSNP-eGene pairs that were significant (replicated) in the replication cohort ($FDR < 0.05$) and had the same direction of effect in both cohorts.” However three of the seven pairs reported in sup table 19 and discussed in the text have a different direction of effect in discovery and replication in table 19, and the table legend even states that :” Two genes with small effect sizes show discordance of effect size in either the replication or the combined datasets”. Opposite directions of effect is a strong indication of false positives and should not be referred to as a replication.

The authors should revise the text, figures and abstract to clearly state that four out of 10 SNP-eGene pairs taken into replication replicated, not seven.

As requested, we modified the text, figures and abstract to clearly state that four eSNP-eGene pairs replicated, and have the same direction of effect in the replication cohort. We also updated Supplemental Figure 15 and 16, and Supplemental table 9 to

only show the four hits that replicate across all cohorts, with the same direction of effect. The main text only describe these 4 findings.

B) I am still confused on how FDR in the replication cohort was calculated. Please explicitly state how the FDR in the replication set was calculated – Sup Figure 15 states “FDR (BH at 0.05) on nb of total genes”. What does nb mean? Does total eGenes here mean all quantified or just those significant in discovery? The authors rebuttal stated “FDR was determined by obtaining q-values for the 10 SNPs tested in the replication cohort. We have added clarification to the relevant section of the main text and the methods.” How do you obtain a q-value for only 10 SNPs? No details have been added to the methods to clarify how FDR in the replication cohort was determined.

We removed the “nb” from Supplemental Figure 15 and changed “BH at 0.05” to “q-value<0.05”. We also clarified the use of FDR assessment using q-values for the discovery and replication cohorts (respectively) in the Materials and Methods and included as a reference the method upon which our calculation of the q-value is based on (Storey, J. D. (2003). "The positive false discovery rate: a Bayesian interpretation and the q-value." Ann. Statist. 31(6): 2013-2035.)”.

7-8) resolved

Finally, given the revised GxE results and the removal of any results directly linking genetic variants to chronic disease in the manuscript, the title of the manuscript “Gene-by-environment interactions in urban populations modulate personal risk to chronic disease” should be modified. This is consistent with the previous request from Reviewer 3 to “If not directly measured (please do if you can), I would recommend tuning down the disease risk statement.” And the authors reply to this request “We have taken this comment into consideration and turned down the disease risk statement.” The authors did indeed tone down the text, but the title still reflects the original submitted manuscript rather than the revised version.

We changed the title to:

Gene-by-environment interactions in urban populations modulate risk phenotypes

REVIEWERS' COMMENTS:

Reviewer #1 (Remarks to the Author):

The authors have sufficiently addressed my points.